# Learning-based Mechanism Design: Truthful, Expressive and Efficient Continuum Approaches for Utility Maximization

## Abstract

Mechanism design is a crucial topic at the intersection of computer science and economics. This paper addresses the automated mechanism design problem by leveraging machine learning and neural networks. The objective is to design a **truthful**, **expressive** and **efficient** mechanism that maximizes the platform's expected utility, given that the players' types are drawn from a pre-specified distribution. We present a general mechanism design model that captures two critical features: hidden information and strategic behavior. Subsequently, we propose the **PFM-Net** framework, which parameterizes the menu mechanism class by function approximation and identifies an optimal mechanism through ingenious optimization techniques. We also provide both theoretical and empirical justifications for the advantages of our approach. Experimental results demonstrate the effectiveness of PFM-Net over traditional and learning-based baselines, enabling the PFM-Net framework to serve as a new paradigm for automated mechanism design.

## 1 Introduction

Designing a truthful mechanism that maximizes the platform's expected utility is a fundamental problem in computational economics, with important application in market design and resource allocation (Börgers et al., 2015; Golowich et al., 2018). In a typical mechanism design problem, the market consists of two kinds of players: platforms (sellers) and customers (buyers), both with given utility functions. The utility functions are determined by the item price, item allocation and the player's own value of items. Typically, the mechanism is required to possess truthfulness (or equivalently, "strategy-proof" & "DSIC and IR") (Likhodedov & Sandholm, 2005) such that the customers have the incentive to report their types honestly and are always willing to participate.

The seminal work of (Myerson, 1981) solved the optimal strategy of selling one item with independent bidder valuations, yet analytical results have been limited to specific simple settings thereafter (Manelli & Vincent, 2006; Giannakopoulos & Koutsoupias, 2014). The machine learning approach to this problem has become the mainstream method, which can be classified into three categories. **VCG-based approaches** (Sandholm & Likhodedov, 2015) define a parameterized class of truthful mechanisms and then optimize within this class. **Regret-based approaches** (Dütting et al., 2019; Ivanov et al., 2022) capture a broad class of mechanisms by incorporating untruthfulness (*i.e.*, regret) as a penalty in the loss function to optimize the mechanism. **Discretization-based approaches** (Duan et al., 2024b; Wang et al., 2024b), including menu-based and mixed integer programming-based methods, discretize the allocation or type space to approximate the optimal mechanism while preserving truthfulness.

Each of these existing approaches has notable drawbacks. **VCG-based methods** are inherently limited in expressive power, making them insufficient to find the optimal mechanism. **Regret-based methods** suffer from untruthfulness, which makes outcomes unpredictable and the mechanism potentially unstable. **Discretization-based approaches** often needs an exponential number of parameters to capture the full type space or allocation space, which becomes prohibitively expensive even for problems with moderate size.

**Our Contributions** In this paper, we close the joint gaps of *truthfulness*, *full expressive power* and *efficiency* in general multi-player mechanism design. We propose a machine learning-based framework called **PFM-Net** (**P**arameterized **F**ull-**M**enu **Net**work) to derive the optimal mechanism.

To achieve this, we first construct a general mechanism design setting in a quasi-linear context, which generalizes auction settings and other scenarios, such as welfare-maximizing platforms. We then characterize truthful mechanisms within this setting, demonstrating that the class of truthful mechanisms is equivalent to the class of menu mechanisms with convex pricing functions, substantially generalizing the results of Rochet (1987) and Hammond (1979).

Building on this characterization, we utilize representations of convex functions, such as **PICNN** (Amos et al., 2017) and **GroupMax** (Warin, 2023), to construct the pricing network. We also derive a training procedure to train the optimal parameterized function, with the objective function formulated as a penalized utility function of the platform. Experimental results validate the effectiveness of PFM-Net framework, by demonstrating that such framework obtain the ability to capture the non-trivial component even in the moderate-sized problems while other methods fail, highlighting its superior performance in moderate-sized problems and its empirical success in avoiding the curse of dimensionality and enabling the PFM-Net framework to serve as a new paradigm for automated mechanism design[1].

## 2 PROBLEM SETTING

**The model** In this paper, we consider a generalized mechanism design model in the quasi-linear context. There are $n$ players, $m$ items as well as **one** platform in this model. Denote $[n] = \{1, 2, ..., n\}$ as the players set and $[m] = \{1, 2, ..., m\}$ as the items set. Each player $i$ has her hidden type $\boldsymbol{t}_i \in \mathcal{T}_i \subseteq \mathbb{R}^m$, and the type space $\mathcal{T}_i$ for player $i$ is public knowledge, assuming to be convex and compact.[2] We denote $\mathcal{T} = \times_{i \in [n]} \mathcal{T}_i$ for simplicity. The $j$'th element of $\boldsymbol{t}_i$, $t_{ij}$, represents the player $i$'s preference for item $j$. Specifically, let $\boldsymbol{x}_i \in \mathbb{R}^m$ be the allocation of items to player $i$. The valuation of the bundle $\boldsymbol{x}_i$ to player $i$ when her type is $\boldsymbol{t}_i$, $v_i(\boldsymbol{x}_i; \boldsymbol{t}_i) = \langle \boldsymbol{t}_i, \boldsymbol{x}_i \rangle + c_i(\boldsymbol{x}_i)$, where $c_i : \mathcal{X}_i \to \mathbb{R}$ is a publicly-known, continuous and differentiable-almost-everywhere regularization term, and $\mathcal{X}_i$ is the feasible allocation set of player $i$. By this form, we only assume that the "hidden part" in valuations, $\langle \boldsymbol{t}_i, \boldsymbol{x}_i \rangle$, is bi-linear on the allocations and hidden types. Note that in this model, the elements in both allocations and types can be positive or negative.[3]

The allocations would bring utilities to the platform as well. Denote $\boldsymbol{x} = \{\boldsymbol{x}_i\}_{i \in [n]}$ and $\boldsymbol{t} = \{\boldsymbol{t}_i\}_{i \in [n]}$ as the allocation profile and type profile of players. We assume no hidden information of the platform, but we allow that the platform's valuation $v_0(\boldsymbol{x}, \boldsymbol{t})$ may depend on type profile $\boldsymbol{t}$, in addition to the allocation profile $\boldsymbol{x}$. Function $v_0(\boldsymbol{x}; \boldsymbol{t})$ is assumed to be continuous and differentiable-almost-everywhere on $\boldsymbol{x}$ as well.

**Quasi-linear utilities** We assume quasi-linear utilities for all players as well as the platform. It means that one-unit of utility can be arbitrarily transformed among players and the platform through one-unit of money paid to or charged from players.[4] Denote $p_i \in \mathbb{R}$ as the payment charged from ($p_i > 0$) or paid to ($p_i < 0$) player $i$, the quasi-linear utilities for player $i$ and platform are,

$$u_i(\boldsymbol{x}_i, p_i; \boldsymbol{t}_i) = v_i(\boldsymbol{x}_i; \boldsymbol{t}_i) - p_i, \ i \in [n], \qquad u_0(\boldsymbol{x}, \boldsymbol{p}; \boldsymbol{t}) = v_0(\boldsymbol{x}; \boldsymbol{t}) + \gamma \sum_{i \in [n]} p_i,$$

where $v_0(\boldsymbol{x}; \boldsymbol{t})$ is the valuation of platform when the allocation is $\boldsymbol{x}$ given the player types $\boldsymbol{t}$, and $\gamma \geq 0$ is a parameter representing how the platform evaluates different outcomes with respect to money and valuations. Both $v_0$ and $\gamma$ are public knowledge, thus excluded from the inputs of $u_0$. The

---

[1] We leave further related works to Appendix A.

[2] A set in Euclidean space is compact if and only if it is closed and bounded.

[3] A positive allocation means the platform allocates the item to the player; while a negative allocation means the platform buys the item from the player. A positive type means the item are "good" for the player such that it increases the valuation of players owing the item; while a negative type means the item are "bad" for player, *e.g.*, pollution, risk and so on.

[4] In the mechanism design literature, quasi-linear utilities and the allowance of money transfer are often indispensable for implementation of truthful mechanisms. (Nisan et al., 2007, §9.3)

formulation of the platform's utility generalizes the social-welfare-oriented platform ($v_0(\boldsymbol{x};\boldsymbol{t}) = \sum_{i\in[n]} v_i(\boldsymbol{x}_i;\boldsymbol{t}_i)$, $\gamma = 0$) or revenue-oriented platform ($v_0(\boldsymbol{x};\boldsymbol{t}) \equiv 0$, $\gamma = 1$), as well as the affine combination of social-welfare and revenue. Throughout this paper, we assume that all players and the platform are expected utility maximizers.

**Allocation constraints**  We allow hard constraints that represent the *feasible allocation set* to each player. Let $\mathcal{X}_i \subseteq \mathbb{R}^m$ be a convex, non-empty set that describes the feasible allocations for player $i$. It means that when the platform assigns allocations $\boldsymbol{x}$ to players, the platform should guarantee that $\boldsymbol{x}_i \in \mathcal{X}_i$, for all $i \in [n]$. $\mathcal{X}_i = \mathbb{R}^m$ means there is no constraint on allocating to player $i$. Denote $\mathcal{X} = \times_{i\in[n]}\mathcal{X}_i$ as the possible allocation set. Note that this model implicitly means that the constraints are endogenous from players, rather exogenous from the platform. [5] We require two technical assumptions: $\boldsymbol{0} \in \mathcal{X}_i$ and $c_i(\boldsymbol{0}) = 0, \forall i$. It means that $\boldsymbol{0}$ is an outside option for all players, with utility normalized to 0. But we also note that these assumptions can be removed without loss of generality.

**Truthful direct mechanisms**  We focus on truthful direct mechanisms in this study. Revelation principle states that focusing on this type of mechanisms is without loss of generalities (Myerson, 1979). In other words, restricting on direct mechanisms do not lose expressiveness. Below we omit the input $(\boldsymbol{t}_1, ..., \boldsymbol{t}_n)$ sometimes when the context is clear. According to convention, $(\boldsymbol{t}_1, ..., \boldsymbol{t}_{i-1}, \boldsymbol{t}'_i, \boldsymbol{t}_{i+1}, ..., \boldsymbol{t}_n)$ is abbreviated as $(\boldsymbol{t}'_i, \boldsymbol{t}_{-i})$ and $(\boldsymbol{t}_1, ..., \boldsymbol{t}_n)$ is abbreviated as $\boldsymbol{t}$. We first present the formal definitions of direct mechanisms for completeness.

**Definition 2.1** (Direct Mechanisms). A direct mechanism $M^d = (\boldsymbol{x}, \boldsymbol{p})$ consists of an allocation rule $\boldsymbol{x} : \mathcal{T} \to \mathcal{X}$ and a payment rule $\boldsymbol{p} : \mathcal{T} \to \mathbb{R}^n$. The mechanism works as follows,

Step 1. The platform requests all players for their types at the same time, and receive the players' report $\boldsymbol{t} = (\boldsymbol{t}_1, ..., \boldsymbol{t}_n) \in \mathcal{T}$.

Step 2. The allocations to players are computed by $\boldsymbol{x}(\boldsymbol{t})$. Each player $i$ is allocated with bundle $\boldsymbol{x}_i$.

Step 3. The payments (or payoffs) of players are computed by $\boldsymbol{p}(\boldsymbol{t})$. Each player $i$ is charged by $p_i$ (or paid $-p_i$) amount of money.

We say a direct mechanism is truthful, if it satisfies two conditions: individual rationality (IR) and incentive compatibility (IC):

$$v_i(\boldsymbol{x}_i(\boldsymbol{t}); \boldsymbol{t}_i) - p_i(\boldsymbol{t}) \geq 0, \qquad\qquad \forall \boldsymbol{t} \in \mathcal{T}, i \in [n] \qquad\qquad \text{(IR)}$$

$$v_i(\boldsymbol{x}_i(\boldsymbol{t}); \boldsymbol{t}_i) - p_i(\boldsymbol{t}) \geq v_i(\boldsymbol{x}_i(\boldsymbol{t}'_i, \boldsymbol{t}_{-i}); \boldsymbol{t}_i) - p_i(\boldsymbol{t}'_i, \boldsymbol{t}_{-i}), \qquad \forall \boldsymbol{t} \in \mathcal{T}, \boldsymbol{t}'_i \in \mathcal{T}_i, i \in [n] \qquad \text{(IC)}$$

The IR condition states that, players are always happy to participate on this mechanism. The RHS in (IR) means that the utility of outside option for each player is $\langle \boldsymbol{t}_i, \boldsymbol{0} \rangle + c_i(\boldsymbol{0}) = 0$. The IC condition states that, truthful telling is a dominant strategy for each player. For simplicity, we abbreviate truthful direct mechanism as *truthful mechanism* later on this paper.

**Our goal**  The goal of this problem is to find a truthful mechanism that maximizes the expected utility of the platform. Regarding the expectation, we assume that the platform holds a prior over the (possibly correlated) joint distribution of players' types, *i.e.*, $\mathcal{F} \in \Delta(\mathcal{T})$. Similar with many learning-based algorithm, we do not require the full knowledge of the distribution $\mathcal{F}$. Instead, the minimum requirement is an access to a sampling oracle, that enables i.i.d. samples of $\{\boldsymbol{t}^k\}_{k\in[K]} \overset{\text{i.i.d.}}{\sim} \mathcal{F}$ with arbitrary sample size $K \geq 1$, which would be utilized by our algorithm.

Formally, the platform's optimization problem is stated as follows.

$$\max_{\substack{\boldsymbol{x}:\mathcal{T}\to\mathcal{X} \\ \boldsymbol{p}:\mathcal{T}\to\mathbb{R}^n}} \mathbb{E}_{\boldsymbol{t}\sim\mathcal{F}} \left[ u_0(\boldsymbol{x}(\boldsymbol{t}), \boldsymbol{p}(\boldsymbol{t}); \boldsymbol{t}) \right] \quad \text{s.t.} \quad (IC), (IR)$$

**Automated mechanism design**  Since the control variables in this problem is infinitely-dimensional[6], finding an analytical optimal solution becomes extremely hard. In this paper, we

---

[5] More discussions about allocation constraints are provided in Appendix F.1

[6] Specifically, the control variables are $\boldsymbol{x}(\cdot)$ and $\boldsymbol{p}(\cdot)$ in mechanism design problem, which are functions on a continuous domain and thus infinitely-dimensional.

follow the framework of *automated mechanism design* (Sandholm, 2003), which parameterizes the mechanism, as a parameterized class, and find the optimal mechanism within this class.

Formally, let $\theta \in \mathbb{R}^{n_\theta}$ be the parameters represented in the allocation rule and payment rule. The mechanism is then represented as $\boldsymbol{x}(\boldsymbol{t}; \theta)$ and $\boldsymbol{p}(\boldsymbol{t}; \theta)$, where $\boldsymbol{x}(\cdot; \cdot)$ and $\boldsymbol{p}(\cdot; \cdot)$ are determined only by network architecture. The problem then reduces to finding the optimal parameter $\theta$,

$$\max_{\theta \in \mathbb{R}^{n_\theta}} \quad \mathbb{E}_{\boldsymbol{t} \sim \mathcal{F}}\left[u_0(\boldsymbol{x}(\boldsymbol{t}; \theta), \boldsymbol{p}(\boldsymbol{t}; \theta); \boldsymbol{t})\right] \quad \text{s.t.} \quad (IC), (IR)$$

**Desirable properties** Before the formal contents, we shall emphasize what desirable properties an ideal approach should possess: **potentially exact truthfulness**, **full expressive power** and **efficiency in moderate-size problem**. More discussions on these properties are presented in Appendix F.2.

## 3 CHARACTERIZATION OF TRUTHFUL MECHANISMS

As is inspired by Wang et al. (2024b) and Dütting et al. (2024), we focus on the menu mechanism class in this paper. We will show that a specific class of menu mechanism characterizes the class of truthful mechanisms in this section. Due to space limits, the formal definition of the menu mechanism is leaved to Appendix B.

For completeness, we briefly introduce *menu mechanisms* in few words. Consider a mechanism design problem with one player, with type space $\mathcal{T}$ and feasible allocation set $\mathcal{X}$. A menu $\mathcal{M}^m = (\mathcal{X}^m, p^m)$ specifies a subset of feasible allocation, $\emptyset \neq \mathcal{X}^m \subseteq \mathcal{X}$, and a pricing rule, $p^m : \mathcal{X}^m \to \mathbb{R}$. The mechanism discloses the menu to the player at first, then the player buy the utility-maximizing allocation $\boldsymbol{x} \in \mathcal{X}^m$ and pay $p^m(\boldsymbol{x})$ money, which depends on her private type $\boldsymbol{t} \in \mathcal{T}$. The case of multi-players is similar. In that case, the platform plays the mechanism with each player independently, with the only difference that the mechanism to each player $i$ can depend on the types of all other players $\boldsymbol{t}_{-i}$.

With a little abuse of notations, we also call $\mathcal{M}^m$ as a menu mechanism with menu $\mathcal{M}^m$. If $\mathcal{X}^m = \mathcal{X}$ always hold, such menu mechanism is called *full-menu mechanism*. Since the properties of *truthfulness* and *full expressiveness* are originally defined for direct mechanisms, it naturally leads to a question that, can we extend such properties to menu mechanisms? Though not intuitive, we shall emphasize that a menu mechanism $\mathcal{M}^m$ can be easily transformed into a direct mechanism $\mathcal{M}^d$. The insight is following: as long as the platform knows the exact types of players, the platform can simulate the player's behaviors as if players' are rationally playing the game. The formal definition is also leaved to Appendix B.

As we can transform each menu mechanism to a direct mechanism, we shall regard them as the "equivalent" mechanism, then consider the properties of menu mechanisms as the properties of "equivalent direct mechanisms". To begin with, we firstly give some definitions that define the equivalence relation between menu mechanisms and direct mechanisms. Note that it's easy to verify below-defined equivalence relation forms an equivalent class in set theory.

**Definition 3.1** (Equivalent mechanisms)**.**

- We say two direct mechanisms $\mathcal{M}_1^d$ and $\mathcal{M}_2^d$ are equivalent, if their allocation rules and payment rules are equal on the domain $\mathcal{T}$, except a set of probability zero. (The probability is measured by $\mathcal{F}$.)

- We say two menu mechanisms $\mathcal{M}_1^m$ and $\mathcal{M}_2^m$ are equivalent, if after we transform $\mathcal{M}_i^m$ into direct mechanisms $\mathcal{M}_i^d$ as above, $\mathcal{M}_1^d$ and $\mathcal{M}_2^d$ are equivalent. [7] We can similarly define equivalent relation between a menu mechanism and a direct mechanism.

- Let $\mathcal{M}^M$ be a class of (direct or menu) mechanisms. Denote $\{-i\} = \{1, 2\} \backslash \{i\}$, we call a pair of mechanism class $\mathcal{M}_1^M$ and $\mathcal{M}_2^M$ are equivalent, if for any $i \in \{1, 2\}$ and any mechanism $\mathcal{M}_i \in \mathcal{M}_i^M$, there is another $\mathcal{M}_{-i} \in \mathcal{M}_{-i}^M$ such that $\mathcal{M}_i$ and $\mathcal{M}_{-i}$ are equivalent.

---

[7] Note that it does not indicate that the pricing functions in $\mathcal{M}_1^m$ and $\mathcal{M}_2^m$ are equivalent, as there can be dummy candidates.

Note that when two mechanism classes are equivalent, these classes have exactly same expressive power. With more abuse of notations, we regard an equivalent class of direct mechanisms or menu mechanisms as same mechanism, denoted by $\mathcal{M}$.

Before the formal statement, we also give some technical definitions that will be used to characterize the mechanism class.

**Definition 3.2** (pricing rule decomposition). Under the situation with one player, allocation constraint $\mathcal{X}$ and regularity cost $c(x)$, we say a *full-menu mechanism* $\mathcal{M}^m = \langle \mathcal{X}, p^m \rangle$ satisfies pricing rule decomposition, if following holds for some $f^m : \mathcal{X} \to \mathbb{R}$,

- $p^m(x) = c(x) + f^m(x)$

- $f^m(x)$ is convex.

Under the situation with $n$ players, allocation constraint $\{\mathcal{X}_i\}_{i \in [n]}$ and regularity cost $\{c_i(x)\}_{i \in [n]}$, we say a *full-menu mechanism* $\mathcal{M}^m = \{\mathcal{M}_i^m\}$ satisfies pricing rule decomposition, if $\mathcal{M}_i^m$ satisfies pricing rule decomposition for all player $i$, whatever $\boldsymbol{t}_{-i}$ is. (Note that $\mathcal{M}_i^m$ may depend on $\boldsymbol{t}_{-i}$.)

**Definition 3.3** (no-buy-no-pay). Under the situation with one player, we say a *full-menu mechanism* $\mathcal{M}^m = \langle \mathcal{X}, p^m \rangle$ satisfies no-buy-no-pay, if $p^m(\boldsymbol{0}) \leq c(\boldsymbol{0}) = 0$.

Under the situation with $n$ player, we say a *full-menu mechanism* $\mathcal{M}^m = \{\mathcal{M}_i^m\}$ satisfies no-buy-no-pay, if $\mathcal{M}_i^m$ satisfies no-buy-no-pay for all player $i$, whatever $\boldsymbol{t}_{-i}$ is.

Now we give a formal statement to show the IC properties for menu mechanisms. Specifically, we have following characterization for these mechanism classes (multi-player version):

**Theorem 3.4.** *Following mechanism classes are equivalent:* [8]

- *The class $\mathcal{M}^{D,IC}$ of direct mechanisms $\mathcal{M}^d = (\boldsymbol{x}, \boldsymbol{p})$ with IC property,*

- *The class $\mathcal{M}^M$ of menu mechanisms $\mathcal{M}^m$, where $\mathcal{M}^m = \{\mathcal{M}_i^m\}_{i \in [n]}$ and $\mathcal{M}_i^m = \{\mathcal{X}_i^m, p_i^m\}$,*

- *The class $\mathcal{M}^{FM,p}$ of full-menu mechanisms $\mathcal{M}^f$, where $\mathcal{M}^f = \{\mathcal{M}_i^f\}_{i \in [n]}$ and $\mathcal{M}_i^f = \{\mathcal{X}_i, p_i^f\}$, satisfying **pricing rule decomposition**.*

The above theorem states that, when we focus on designing IC mechanisms, restricting mechanism within the menu mechanism class $\mathcal{M}^M$ (or full-menu mechanism class with pricing rule decomposition, $\mathcal{M}^{FM,p}$) is without loss of generality. Next we will show that the IR constraints can be resolved in a similar way.

**Theorem 3.5.** *Following mechanism classes are equivalent:*

- *The class $\mathcal{M}^{D,T}$ of truthful direct mechanisms $\mathcal{M}^d = (\boldsymbol{x}, \boldsymbol{p})$ (IC & IR),*

- *The class $\mathcal{M}^{FM,pn}$ of full-menu mechanisms $\mathcal{M}^f$, where $\mathcal{M}^f = \{\mathcal{M}_i^f\}_{i \in [n]}$ and $\mathcal{M}_i^f = \{\mathcal{X}_i, p_i^f\}$, satisfying **pricing rule decomposition** and **no-buy-no-pay**.*

# 4 METHODOLOGY

Inspired by Theorem 3.5, we only need to find the optimal mechanism within the mechanism class $\mathcal{M}^{FM,pn}$, without considering truthfulness constraints. The property of full expressive power has also been preserved.

**Mechanism representation** The only degrees of freedom in $\mathcal{M}^{FM,pn}$ lies in the flexible pricing rule. We begin with parameterizing the pricing rule (*i.e.*, parameterizing the full-menu mechanism). Denote $\mathcal{M}^{PFM}$ as the class of **P**arameterized **F**ull-**M**enu mechanisms, $\Theta$ as the set of parameters to parameterize this class (*e.g.*, weights and bias in a neural network) and $\theta \in \Theta$ as a parameter

---

[8] Hammond (1979) derived the relation between IC mechanism and menu mechanism, while Rochet (1987) derived the convex utility function in truthful mechanism, we argue that our characterization results are different from theirs and in fact more general. See Appendix A for more details.

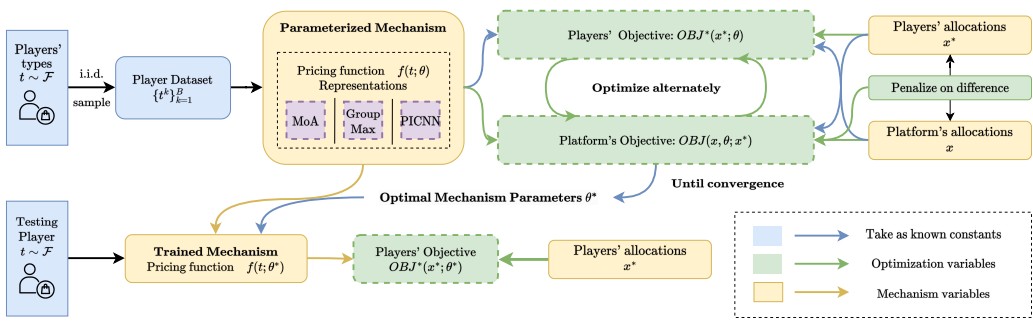

Figure 1: The overview of our algorithm. In the training process, we first sample a sufficiently large data set from the given player type distribution. Our characterization demonstrates the pricing function $f$ to be convex, therefore a representation of convex function is chosen to express $f$. We train the mechanism by alternately optimizing the platform and players' objective function, while gradually increasing the penalty of difference between the two allocation matrices to reach platform-player consensus (which represents the full mechanism) and the convergence of parameter optimization. In the testing step, we fix the near-optimal mechanism parameters $\theta^*$ and test the sampled players utilities as the final result.

instance. Specifically, the pricing rule is parameterized as follows,

$$p_i(\boldsymbol{x}_i; \boldsymbol{t}_{-i}; \theta) = c_i(\boldsymbol{x}_i) + f_i(\boldsymbol{x}_i; \boldsymbol{t}_{-i}; \theta)$$

By *pricing rule decomposition*, we know that an optimal $f_i(\boldsymbol{x}_i; \boldsymbol{t}_{-i}; \theta)$ should be convex on $\boldsymbol{x}_i$ within $\mathcal{M}^{FM,pn}$, therefore, we also restrict $f_i(\boldsymbol{x}_i; \boldsymbol{t}_{-i}; \theta)$ to be convex within $\mathcal{M}^{PFM}$.

To do this, we need an expressive convex representation of convex function class. There are many such options for this goal. We implement maximum-of affine functions (MoA, Balázs et al. (2015)), log-sum-exp functions (LSE, Kim & Kim (2022)), Partial Input Convex Neural Networks (PICNN, Amos et al. (2017)), Group Max neural networks (GroupMax, Warin (2023)). See more details in Appendix G.2.

Notice that *no-buy-no-pay* property requires that $f_i(\boldsymbol{0}; \boldsymbol{t}_{-i}; \theta) \leq 0$. To resolve this requirement, we hard-code this constraint within $\mathcal{M}^{PFM}$. A general way is to replace $f_i(\boldsymbol{x}_i; \boldsymbol{t}_{-i}; \theta)$ with

$$\hat{f}_i(\boldsymbol{x}_i; \boldsymbol{t}_{-i}; \theta) = f_i(\boldsymbol{x}_i; \boldsymbol{t}_{-i}; \theta) - f_i(\boldsymbol{0}; \boldsymbol{t}_{-i}; \theta),$$

where the second term in RHS represents a normalization constant. We can easily verify that $\hat{f}_i(\boldsymbol{0}; \boldsymbol{t}_{-i}; \theta) = 0$. Other hard-coding approaches for specific models are represented in appendix Appendix G.2.

**Learning-based algorithm**   We leave the derivations of our algorithm to Appendix E. Figure 1 briefly present the procedure of our algorithm (both training and inference).

**Real-time inference**   After learning the mechanism $\theta^*$, the ultimate goal for this mechanism is to operate effectively on an unseen type profile $\boldsymbol{t}$. To achieve this, we can directly compute the utility-maximizing allocations for each player $i$ by optimizing her utility: $\boldsymbol{x}_i^* \in \arg\max_{\boldsymbol{x}_i \in \mathcal{X}_i} u_i(\boldsymbol{x}_i; p_i(\boldsymbol{x}_i; \boldsymbol{t}_{-i}; \theta^*); \boldsymbol{t}_i)$, and charge payment $p_i(\boldsymbol{x}_i^*; \boldsymbol{t}_{-i}; \theta^*)$ for player $i$.

## 5   JUSTIFICATION OF PFM-NET

In this section, we justify the advantages of PFM-Net both theoretically and empirically.

**Truthfulness.**   The truthfulness of $\mathcal{M}^{PFM}$ is a direct corollary of Theorem 3.5. as $\mathcal{M}^{PFM} \subseteq \mathcal{M}^{FM,pn}$.

**Universal approximation properties.** We show the universal approximation property of $\mathcal{M}^{PFM}$ in this part. We leave the formal definition of universal approximators to Appendix B.2.

Kim & Kim (2022) studied the universal approximator properties of parameterized MoA functions for approximating convex functions. A straightforward argument shows that even we restricting $f(0) \leq 0$, the universal approximation property remains valid:

**Proposition 5.1.** *The mechanism class $\mathcal{M}^{PFM}$ is a universal approximator for the mechanism class $\mathcal{M}^{FM,pn}$, if the pricing functions are represented by MoA, LSE, GroupMax, or GroupLSE.*

However, what truly concerns us is not the convex pricing function itself, but the expected utility of the platform. Next, we demonstrate that if the full-menu mechanism class $\mathcal{M}^1$ is a universal approximator for another full-menu mechanism class $\mathcal{M}$ under the $L_\infty$ norm, then the expected utility retains the universal approximation property as well. We begin with some technical definitions.

**Definition 5.2** (Non-degenerate distribution). Let $\mathcal{X}$ be a full-dimensional subspace of $\mathbb{R}^d$. We say that $\mathcal{D}$ is a non-degenerate distribution over $\mathcal{X}$, if for any subset $\mathcal{X}_0 \subseteq \mathcal{X}$, we have $\Pr_{x \sim \mathcal{D}}[x \in \mathcal{X}_0] > 0$ indicates that $\mu(\mathcal{X}_0) > 0$, where $\mu(\cdot)$ is Lebesgue measure.

**Definition 5.3** (Maximum expected utility). Let $\mathcal{M}$ be a mechanism class represented by convex function $p(\boldsymbol{x}; \boldsymbol{t}) \in \mathcal{M}$, $\mathcal{T}$ and $\mathcal{X}$ are compact subset of Euclidean spaces $\mathbb{R}^d$, $\boldsymbol{t} \sim \mathcal{D}$ be some non-degenerate distribution over $\mathcal{T}$, and $v_0(\boldsymbol{x}; \boldsymbol{t})$, $\lambda \geq 0$ are the valuation and the quasi-linear parameter of the platform. We define the maximal expected utility within class $\mathcal{M}$, MEU($\mathcal{M}$), as follows,

$$\text{MEU}(\mathcal{M}) = \sup_{\substack{p \in \mathcal{M} \\ x:\mathcal{T} \to \mathcal{X}}} \mathbb{E}_{t \sim \mathcal{D}} \left[ v_0(x(t); t) + \lambda \cdot p(x(t); t) \right] \tag{1}$$

subject to the constraints:

$$x(t) \in \arg\max_{x \in \mathcal{X}} \langle x, t \rangle - p(x; t), \quad \forall t \in \mathcal{T}$$

The formal statement is follows:

**Theorem 5.4.** *Assume that $\mathcal{M}^1$ is a universal approximator of $\mathcal{M}$ under following technical conditions,*

1. *$v_0(x; t)$ is continuous on $\mathcal{X}$ and $\mathcal{T}$ (thus continuous consistently).*

2. *The pricing function $p(x, t)$ is $\varepsilon_1$-strongly convex on $x$ for some $\varepsilon_1 > 0$, when $p \in \mathcal{M}$.*

*Then, $\text{MEU}(\mathcal{M}^1) = \text{MEU}(\mathcal{M})$.*

Theorem 5.4 indicates that using convex representations such as MoA, LSE, GroupMax, and GroupLSE does not result in any loss of expected utility of platform, since the objective of mechanism design problem is a specific form of Equation (1). Although we assume the strong convexity of the optimal pricing rule $\boldsymbol{p}(\boldsymbol{x}_i; \boldsymbol{t}_{-i})$, we note that this is only a technical condition, which is not strong because $\varepsilon_1$ can be chosen so small that strong convex function can be arbitrary close to any convex function in bounded domain. We believe that the theorem also holds even if this condition is moved.

**Efficiency in expressive power** In this section, we examine whether a reasonable number of parameters can approximate a wide range of full-menu mechanisms with a small error. It is clear that the entire class of convex functions can not be fully approximated well by polynomial number of parameters and suffer from curse of dimensionality inevitably with smoothness prior only (Bengio et al., 2005).

Thus, we shift to an alternative solution concept. We argue that our method could practically exhibit greater expressive power compared to existing approaches. We compare our approaches to discretization-based methods (*e.g.*, Wang et al. (2024b)) and AMA-based methods (*e.g.*, Curry et al. (2023)).

COMPARISON WITH DISCRETIZATION-BASED METHODS It is widely believed that realistic high-dimensional problems often exhibit favorable structures that can be effectively captured using sub-exponential numbers of parameters. One promising approach is to leverage network structures, and neural networks are commonly regarded as an ideal tool for approximating high-dimensional functions.

Our methods utilize the PICNN and GroupMax network architectures to approximate the pricing function. However, it remains unclear how to effectively combine network architectures with discretization-based approaches.[9] Without the flexibility of network architectures, discretization-based approaches are particularly susceptible to the curse of dimensionality (Bellman, 1966).

COMPARISON WITH AMA-BASED METHODS    An AMA mechanism is determined by positive weights $\boldsymbol{w} = (w_1, ..., w_n) \in \mathbb{R}_+^n$ of players as well as a shift function $\lambda(x)$ on allocations. The formal definition of how AMA mechanism would work in our model is leaved to Appendix B.3. We have following comparison:

**Proposition 5.5.** *Consider an AMA mechanism $M^{AMA}$ with positive weights $w_1, \ldots, w_n$ and a shift function $\lambda(\boldsymbol{x})$. Assume more that an oracle $\mathcal{O}^{AMA}$ of AMA mechanism exists that can run the mechanism $M^{AMA}$ under input $\boldsymbol{t}$. Formally, $\mathcal{O}^{AMA}$ receives $M^{AMA}$ (or equivalently, $\boldsymbol{w}$ and $\lambda$) and $\boldsymbol{t}$ as inputs, and output the resulting allocation $\boldsymbol{x}$ and payment $\boldsymbol{p}$.*

*Given any AMA mechanism $M^{AMA}$, we can explicitly construct a full-menu mechanism $M^F$ with pricing functions $\{p_i^f(\boldsymbol{x}_i; \boldsymbol{t}_{-i})\}_{i \in [n]}$, that receives type profile $\boldsymbol{t}$, outputs the full menu $p_i : \mathcal{X}_i \to \mathbb{R}, i \in [n]$, and is equivalent to $M^{AMA}$.*

*Additionally, querying $\{p_i(\boldsymbol{x}_i)\}_{i \in [n]}$ at some point $\boldsymbol{x} \in \mathcal{X}$ needs polynomial-time computation and $O(n)$ black-box queries of the oracle $\mathcal{O}^{AMA}$.*

Proposition 5.5 states that, our framework can efficiently simulate AMA mechanisms.

In the reverse direction, it is well-known that the AMA mechanism class lacks full expressive power (Carbajal et al., 2013). Given that PFM-Net exhibits full expressive power, there must exist an instance of PFM-Net that cannot be expressed by an AMA mechanism.

# 6    EXPERIMENTS

In this section, we conduct empirical experiments that evaluate the effectiveness of PFM-Net. The pricing functions are parameterized by MoA , PICNN (Amos et al., 2017) and GroupMax (Warin, 2023).

## 6.1    BASELINES METHODS

We present the manually defined baselines and learning-based baselines we compared in this part. The **manually defined baselines** include,

1. **VCG** (Vickrey, 1961): The most classical mechanism with strong versatility.
2. **Item-wise Myerson**: Item-wise Myerson is a auction baseline used in Dütting et al. (2019), that sells the $m$ items independently and optimally to the players.
3. **Bundle-OPT**: this mechanism bundles all items together at a specific price when selling items to buyers. The price is parameterized, and the optimal price is selected for each setting by one-dimensional grid search. This mechanism is particularly effective when there is only one player in the game. This baseline is also used in Curry et al. (2023).

The **learning-based baselines** include:

1. **Lottery-AMA** (Curry et al., 2023): Lottery-AMA is an AMA-based approach that sets bidder weights, along with the discretization of the allocation menu and shift values, as learnable parameters. We also made appropriate extensions to fit it into our actual experimental setting.
2. **UM-GemNet**: An extension of GemNet (Wang et al., 2024b) that can fit our generalized mechanism design setting. GemNet is a menu-based approach that discretes the menu for each bidder, which is computed by a fully-connected neural network taking others' types as input. [10]

---

[9]Although GemNet incorporates a network, we emphasize that their network is solely used to output a set of allocation points to form the menu. The menu itself inherently discretizes the allocation space.

[10]We point out that in original implementation of GemNet, there is an integer-programming based transformation after the training of GemNet, which is used to transform GemNet such that it's menu compatible. We do not incorporate this transformation in our implementations of both UM-GemNet and PFM-Net.

Table 1: The experimental results of selling multiple goods to one buyer. The distribution $t \sim U([0,1]^m)$. $S_m$ represents the experiments of selling $m$ goods. The values represent the expected utility of the seller, with the maximum value on bold.

| Methods & Settings | $S_2$ | $S_3$ | $S_5$ | $S_{10}$ | $S_{15}$ | $S_{20}$ |
|---|---|---|---|---|---|---|
| PICNN-1 | 0.5472 | 0.8695 | 1.5740 | 3.4527 | 5.4444 | 7.5291 |
| GroupMax-1 | **0.5476** | **0.8751** | 1.5746 | 3.4568 | 5.4567 | 7.5784 |
| GroupMax-3 | 0.5468 | 0.8705 | **1.5774** | **3.4838** | **5.5525** | **7.6225** |
| UM-GemNet | 0.5442 | 0.8726 | 1.5560 | 3.4411 | 5.4284 | 7.5167 |
| Lottery-AMA | 0.5402 | 0.7952 | 1.0932 | - | - | - |
| Item-wise Myerson | 0.5000 | 0.7500 | 1.2500 | 2.5000 | 3.7500 | 5.0000 |
| Bundle-OPT | 0.5441 | 0.8599 | 1.5557 | 3.4491 | 5.4543 | 7.5290 |
| OPT | 0.5491 | 0.8757 | - | - | - | - |

Note that all of these baseline models were originally implemented in the context of auction settings. In our experiments, we made slight modifications to the implementations of lottery-AMA and GemNet to ensure their applicability to scenarios that extend beyond traditional auction problems.

## 6.2 EXPERIMENTAL SETTINGS

### 6.2.1 SELLING TO SINGLE BUYER

In this experiment, we consider the problem of selling $m$ items to a single buyer. The bidder's type distribution is $t$ i.i.d. $U([0,1]^m)$. The buyer has an allocation constraint of $\mathcal{X} = [0,1]^m$, meaning that the quantity of each item purchased cannot exceed 1. Both the buyer and the platform have no intrinsic valuation for the allocations, i.e., $v_0(\boldsymbol{x}) = c_1(\boldsymbol{x}) = 0, \forall \boldsymbol{x}$. Therefore, the platform's expected utility is equivalent to its expected revenue. We denote $S_m$ as the problem involving $m$ items in this setting.

We implement MoA, 1-layer PICNN, 1-layer GroupMax, and 3-layer GroupMax architectures within PFM-Net. As baselines, we also implement UM-GemNet and lottery-AMA as learning-based baselines, alongside two simple baselines: item-wise Myerson and Bundle-OPT. We compare the performance of these methods for $m = 2, 3, 5, 10, 15, 20$.

The expected revenues for different settings are presented in Table 1, with the optimal value for each setting highlighted in bold. Note that optimal values (OPT) have only been found in special cases, namely for two or three items by Manelli & Vincent (2006). The OPT for two items is computed analytically, while for three items it is computed numerically with random $1,000,000$ samples.

The MoA-based PFM-Net and lottery-AMA do not perform well for larger-scale problems, so some results are omitted.

### 6.2.2 SOCIAL PLANNER OF A MARKET

In this experiment, we consider the problem faced by a social planner aiming to maximize social welfare by designing a market. Let there be $n$ agents and $m$ goods in a market. The agents' types are generated independently and identically distributed (i.i.d.) from either a uniform distribution $U([-1,1])$ or a normal distribution $\mathcal{N}(0,1)$. We denote $P_{n,m}^F$ as the problem with $n$ agents and $m$ goods, where the types are i.i.d. from distribution $F$. Specifically, $F = U$ represents the uniform distribution, and $F = N$ represents the normal distribution.

We set the allocation constraint for each agent as $\mathcal{X}_i = [-1,1]^m$, indicating that each agent can either buy or sell the goods in the market, with a maximum amount of 1. We incorporate a regularity term to describe diminishing marginal utility, i.e., each agent has a regularization term $c_i(\boldsymbol{x}) = -\frac{1}{2}\|\boldsymbol{x}\|^2$ for allocation $\boldsymbol{x}$. Specifically, the utility of agent $i$ is given by:

$$u_i(\boldsymbol{x}_i; \boldsymbol{t}_i; p_i) = v_i(\boldsymbol{x}_i; \boldsymbol{t}_i) - p_i, \qquad v_i(\boldsymbol{x}_i; \boldsymbol{t}_i) = \langle \boldsymbol{x}_i, \boldsymbol{t}_i \rangle - \frac{1}{2}\|\boldsymbol{x}_i\|^2$$

Table 2: The experimental results of social planners in a market. $P_{n,m}^F$ represents a society with $n$ agents, $m$ items and types are i.i.d. distributed from distribution $F$. $F = N$ represents normal distribution with mean $0$ and standard deviation $1$, and $F = U$ represents uniform distribution in $[-1, 1]$. Allocation constraints of players are $\mathcal{X}_i = [-1, 1]^m$. The utility of the platform is the social welfare, minus a penalty capturing the disobey of market clearance. The values represent the expected utility of the social planner, with the maximum value on bold.

| Methods & Settings | $P_{1,5}^U$ | $P_{1,5}^N$ | $P_{2,5}^U$ | $P_{2,5}^N$ | $P_{3,5}^U$ | $P_{3,5}^N$ |
|---|---|---|---|---|---|---|
| GroupMax-1 | **0.3853** | **1.1399** | **1.0165** | **2.6812** | **1.6512** | **4.2900** |
| UM-GemNet | 0.3261 | 0.9013 | 0.8949 | 2.4251 | 1.4367 | 3.7948 |
| VCG | 0 | 0.8603 | 0 | 1.7188 | 0 | 2.5846 |
| OPT | 0.4167 | 1.2348 | 1.1101 | - | - | - |

The social planner is oriented towards maximizing social welfare and therefore has no direct utility over monetary exchanges. The market must also satisfy the market clearance condition, which requires that the total quantity of each item remains unchanged before and after the mechanism. In our model, we assume that the social planner incurs a quadratic cost for any violation of the market clearance condition. Specifically, the utility of the social planner is:

$$u_0(\boldsymbol{x}; \boldsymbol{t}; \boldsymbol{p}) = \sum_{i=1}^n v_i(\boldsymbol{x}_i; \boldsymbol{t}_i) - \frac{1}{2} \sum_{j=1}^m \left( \sum_{i=1}^n x_{ij} \right)^2$$

where the term $\frac{1}{2} \left( \sum_{i=1}^n x_{ij} \right)^2$ represents the platform's effort cost when the total surplus or demand of item $j$ is $\sum_{i=1}^n x_{ij}$.

We compare the performance of 1-layer GroupMax, GemNet, and VCG in settings with 5 items and 1, 2, or 3 players, under both uniform and normal distribution assumptions. The expected utilities for the different settings are presented in Table 2, with the optimal value for each setting highlighted in bold.

### 6.3 EXPERIMENTAL ANALYSIS

**Selling to single buyer**   We find that the performance of all methods exceeds the strong baseline of Bundle-OPT when $m \le 5$, except for lottery-AMA. This is not surprising, as Bundle-OPT involves only a single parameter, making it an easy baseline to learn. In the case of $m = 2$, these methods also nearly approach the optimal mechanism. However, when $m \ge 5$, we observe that UM-GemNet performs very similarly to Bundle-OPT. In comparison, the 3-layer GroupMax significantly outperforms both UM-GemNet and Bundle-OPT, suggesting that the GroupMax network learns some nontrivial components beyond the simple mechanism of selling the full bundle at a fixed price, whereas UM-GemNet does not. These findings support our conjecture that UM-GemNet, as well as other discretization-based methods, are vulnerable to problems of moderate size. More in-depth analysis of the "non-trivial components" in the learned pricing rule is provided in Appendix G.3.

**Social planner of a market**   The performance of GroupMax exceeds that of GemNet and VCG across all settings. We derive the optimal solution (OPT) in cases where the analytical optimal solution exists. Additionally, we find that the value with $n \ge 2$ players is greater than $n$ times the value with a single player, except in the case of VCG. This observation is due to the insight that if one player wants to buy an item (i.e., $t > 0$), and another player is willing to sell it (i.e., $t < 0$), they can reach an agreement that enhances social welfare. Specifically, in all scenarios, the value obtained by PFM-Net with $n$ players exceeds $n$ times the optimal value achieved with a single player. In a demonstration of the pricing rule of GroupMax in Appendix G.3, we randomly selected three type profiles and examined the learned pricing rule for player 1. We observed that the pricing rule significantly changes with the types of other players, indicating that PFM-Net successfully learns a conditional pricing rule based on the other players' types.[11]

---

[11]A more detailed analysis of both experiments is provided in Appendix G.3.

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

# APPENDIX

## A  FURTHER RELATED WORKS

**Automated Mechanism Design**  Automated mechanism design was first proposed by Sandholm (2003), with various applications including ad auctions (Benisch et al., 2009), combinatorial auctions (Sandholm & Likhodedov, 2015), and mechanism design without money (Narasimhan et al., 2016). Balcan et al. (2016) explored the sample complexity of automated mechanism design, while Albert et al. (2017) investigated the robust automated mechanism design problem. Additionally, Zhang & Conitzer (2021) studied dynamic automated mechanism design.

**Differential Economics for Mechanism Design**  Differential economics aims at parameterizing differential functions to represent the optimal economic solutions, and find the optimal solution from gradient, which can be seen as a sub-field of automated mechanism design (Wang et al., 2024a). Shen et al. (2018) uses the neural network to assist the mechanism design problem. Dütting et al. (2019) begins with optimal auction design by auction. Ivanov et al. (2022) incorporates transformer architecture into auction design. Curry et al. (2023) proposes lottery-AMA, which discrete the platform's allocation space and optimize an AMA mechanism. Duan et al. (2024b) later studies the optimal combinatorial auction design within VVCG mechanism class. The most related works with us should be Wang et al. (2024b), which studies the menu mechanism, with the menu depends on other players types. The menu discretes each player's allocation space.

**Characterization of Truthful Mechanisms**  Hammond (1979) showed that menu-based mechanism is a sufficient condition for IC, and IC mechanism is in some sense a menu mechanism. Rochet (1987) showed that an truthful mechanism will induce the utility of player convex on her type. Though similar to our results, we argue that our characterization in Section 3 is different from theirs and more general. Compared with Hammond (1979), they mainly focus on the discrete menu mechanism, and their results has no convexity characterization. Our characterization show the nature of full-menu and convexity, making the truthful mechanisms and convex full-menu mechanisms "equivalent class" rather than sufficient and necessary condition, making the characterizations of Hammond (1979) more concise. Compared with Rochet (1987), they studies the convex utility of truthful mechanism, having no connection on menu mechanism. Our results state that, if we transform a truthful mechanism into menu mechanism, the pricing rule of the menu is also convex. These two results have different perspectives, which are complementary to each other.

There are also a plenty of works characterized the relation between truthful mechanisms and VCG-based mechanisms. Roberts (1979) shows that if the valuation spaces are full domain and player number is no less than 3, then any implementable mechanism must be an AMA. A simplified proof is later provided by Lavi et al. (2009). In a general setting of combinatorial auctions, Lavi et al. (2003) proves that any implementable mechanism should be "almost" AMA. All of their works studies deterministic mechanism. To the best of our knowledge, there are no full characterization about the general class of randomized mechanism.

**Characterization of Optimal Auctions**  Since the seminal work of Myerson (1981) for optimally selling one item to independent buyers, there are only special cases that optimal solution has been found over the past 40 years (Manelli & Vincent, 2006; Pavlov, 2011; Giannakopoulos & Koutsoupias, 2014; Daskalakis et al., 2015; Yao, 2017) Among these, Manelli & Vincent (2006) studies bundling mechanism, showing the condition such that bundling mechanism is optimal among all truthful mechanisms and deriving the optimal mechanism in selling 2 or 3 uniform items. Giannakopoulos & Koutsoupias (2014) generalizes the results to up $to$ 6 uniform items, though the optimal mechanism is not analytically given. Yao (2017) studies the optimal mechanism when there are two items with discrete distributions.

**Representing Convex Functions**  Max-of-Affine (MoA) functions and Log-sum-exp (LSE) functions are well-known functions that are convex by design. Calafiore et al. (2019) demonstrates that both the maximum-of-affine and log-sum-exp functions are universal approximators for the class of convex functions under the $L_\infty$ norm. Later, Kim & Kim (2022) further shows that conditioned maximum-of-affine and conditioned log-sum-exp are also universal approximators for the class of continuous functions that exhibit convexity over partial inputs. Additionally, Warin (2023) proves

that GroupMax can represent the maximum-of-affine function[12], making both GroupMax and GroupLSE universal approximators as well. Partial input convex neural network (PICNN) has been proposed by Amos et al. (2017) to represent neural-network based convex functions. However, to the best of our knowledge, there are no established results confirming whether ICNN or PICNN is a universal approximator.

# B    Supplementary Definitions

## B.1    Menu Mechanisms

We introduce the menu mechanism with one player first, then extend menu mechanisms to multiple players.

**Definition B.1** (Menu mechanism with **one** player).
Consider a mechanism design problem with one player, with type space $\mathcal{T}$ and feasible allocation set $\mathcal{X}$. A menu $\mathcal{M}^m = (\mathcal{X}^m, p^m)$ specifies a subset of feasible allocation, $\emptyset \neq \mathcal{X}^m \subseteq \mathcal{X}$, and a pricing rule, $p^m : \mathcal{X}^m \to \mathbb{R}$. $p^m(x^m)$ means that the player will pay $p^m(x^m)$ to get the bundle $x^m \in \mathcal{X}^m$. Note that the menu does not depend on the hidden type of the player. The mechanism works as follows.

Step 1. The platform presents the menu $\mathcal{M}^m$ to the player.

Step 2. After seeing the menu $\mathcal{M}^m$, the utility-maximizing player with type $t \in \mathcal{T}$ choose the bundle $x^*(t)$ that maximizes her quasi-linear utility and report $x^*(t)$ to the platform. Specifically,

$$x^*(t) \in \arg\max_{x^m \in \mathcal{X}^m} u(x^m, p^m(x^m); t) \tag{2}$$

Step 3. The player and the platform reach a deal of $x^*(t)$. The player need to pay $p^m(x^*(t))$ to the platform.

**Definition B.2** (Menu mechanism with **multiple** players).

Consider a mechanism design problem with $n$ players, with type space $\mathcal{T} = \times_{i \in [n]} \mathcal{T}_i$ and feasible allocation set $\mathcal{X} = \times_{i \in [n]} \mathcal{X}_i$. The mechanism works as follows:

Step 1. The mechanism requests the type profile of players $\boldsymbol{t}$.

Step 2. For each player $i$, the mechanism construct a menu $\mathcal{M}_i^m$ for player $i$, given $\boldsymbol{t}_{-i}$.

Step 3. For each player $i$, the mechanism runs the one-player mechanism with menu $\mathcal{M}_i^m$ and player $i$.

Specifically, the mechanism defines $n$ conditional menu functions $\mathcal{X}_i^m : \mathcal{T}_{-i} \to \mathcal{P}(\mathcal{X}_i)$ as well as $n$ conditional pricing functions $p_i^m : \mathcal{X}_i^m(\boldsymbol{t}_{-i}) \times \mathcal{T}_{-i} \to \mathbb{R}$. The player $i$ will choose

$$x_i^*(t_i; \boldsymbol{t}_{-i}) \in \arg\max_{x_i^m \in \mathcal{X}_i^m(\boldsymbol{t}_{-i})} u(x_i^m, p_i^m(x_i^m; \boldsymbol{t}_{-i}); t_i)$$

## B.2    Universal Approximators

We show the definition of universal approximators in this section.

**Definition B.3** ($L_\infty$ norms between full-menu mechanisms). Let $M_1^{FM}$ and $M_2^{FM}$ be two full-menu mechanisms. We denote the $L_\infty$ norm as follows (note that full-menu mechanisms are only

---

[12]It is straightforward to see that GroupLSE can represent the log-sum-exp (LSE) function, although Warin (2023) does not provide explicit results.

represented by pricing functions),

$$l_\infty(M_1^{FM}, M_2^{FM}) = \max_{i \in [n]} \sup_{\boldsymbol{t}_{-i} \in \mathcal{T}_{-i}} \sup_{\boldsymbol{x}_i \in \mathcal{X}_i} |p_1(\boldsymbol{x}_i; \boldsymbol{t}_{-i}) - p_2(\boldsymbol{x}_i; \boldsymbol{t}_{-i})|$$

$$= \max_{i \in [n]} \sup_{\boldsymbol{t}_{-i} \in \mathcal{T}_{-i}} \sup_{\boldsymbol{x}_i \in \mathcal{X}_i} |f_1(\boldsymbol{x}_i; \boldsymbol{t}_{-i}) - f_2(\boldsymbol{x}_i; \boldsymbol{t}_{-i})|$$

, where $p_1(\boldsymbol{x}_i; \boldsymbol{t}_{-i}), p_2(\boldsymbol{x}_i; \boldsymbol{t}_{-i})$ are pricing functions of $\mathcal{M}_1^{FM}, \mathcal{M}_2^{FM}$, respectively.

**Definition B.4** (Universal Approximator). We call full-menu mechanism class $\mathcal{M}_1$ a universal approximator of another full-menu mechanism class $\mathcal{M}$. If following two conditions hold,

1. $\mathcal{M}_1 \subseteq \mathcal{M}$.

2. Given any $M \in \mathcal{M}$ and any $\varepsilon > 0$, we can find a full-menu mechanism $M_1 \in \mathcal{M}_1$ such that $l_\infty(M_1, M) < \varepsilon$.

### B.3 AFFINE MAXIMIZER MECHANISMS

We extend the mechanism of affine maximizer auction to fit our model in this section. We call such mechanism as *Affine Maximizer Mechanism* (AMM).

**Definition B.5** (Affine Maximizer Mechanism). Denote $w_0 \in \mathbb{R}, w_1, ..., w_n \in \mathbb{R}_+^n$ as the weights of players and $\lambda : \mathcal{X} \to \mathbb{R}$ as the offset of the allocation. Define the affine social welfare of allocation $\boldsymbol{x}$ and type profile $\boldsymbol{t}$ as follows,

$$\text{ASW}(\boldsymbol{x}; \boldsymbol{t}) = \sum_{i \in [n]} w_i v_i(\boldsymbol{x}_i; \boldsymbol{t}_i) + \lambda(\boldsymbol{x})$$

$$\text{ASW}_{-i}(\boldsymbol{x}; \boldsymbol{t}_{-i}) = \sum_{j \neq i} w_j v_j(\boldsymbol{x}_j; \boldsymbol{t}_j) + \lambda(\boldsymbol{x})$$

The affine maximizer mechanism works as follows,

Step 1. The mechanism requests the players' type profile $\boldsymbol{t}$.

Step 2. Compute $\boldsymbol{x}^* \in \arg\max_{\boldsymbol{x} \in \mathcal{X}} \text{ASW}(\boldsymbol{x}; \boldsymbol{t})$. Take $\boldsymbol{x}^*$ as true allocation.

Step 3. For each player $i$, find $\boldsymbol{t}_i^*$ such that $\max_{\boldsymbol{x} \in \mathcal{X}} \text{ASW}(\boldsymbol{x}; \boldsymbol{t}_i^*, \boldsymbol{t}_{-i})$ get minimum. Denote the corresponding allocation as $\boldsymbol{x}^{-i}$. Take $\boldsymbol{x}^{-i}$ as virtual allocation without the participation of player $i$.

Step 4. For each player $i$, compute $p_i = \frac{1}{w_i} \left[ \text{ASW}(\boldsymbol{x}^{-i}; \boldsymbol{t}_i^*, \boldsymbol{t}_{-i}) - \text{ASW}_{-i}(\boldsymbol{x}^*; \boldsymbol{t}) \right]$.

Step 5. For each player $i$, allocate $\boldsymbol{x}_i^*$ to player $i$ and charge her $p_i$ money.

## C SUPPLEMENTARY LEMMA

### C.1 A SIMPLIFIED VERSION OF ENVELOPE THEOREM

**Lemma C.1** (Envelope Theorem (Milgrom & Segal, 2002)). *Let $f(x, y)$ be a differential function and $y^*(x) = \arg\max_y f(x, y)$. Denote $g(x) = f(x, y^*(x))$ then,*

$$\frac{\partial g}{\partial x}(x) = \frac{\partial f}{\partial x}(x, y)|_{y=y^*(x)}$$

*Proof of Lemma C.1.* We know that by integration of nested functions,

$$\frac{\partial g}{\partial x}(x) = \frac{\partial f}{\partial x}(x, y^*(x)) + \frac{\partial f}{\partial y}(x, y^*(x))\frac{\partial y^*}{\partial x}(x)$$

By argmax property of $y^*(x)$, we know that

$$\frac{\partial f}{\partial y}(x, y^*(x)) = 0$$

which follows the original equality.

$\square$

This lemma tells that, when we want to compute $\frac{\partial g}{\partial x}(x)$, we do not need to compute $\frac{\partial y^*}{\partial x}(x)$. We only need to compute $y^*(x)$.

## C.2 TRUTHFULNESS OF AMM

**Lemma C.2** (Truthfulness of AMM). *The extended affine maximizer mechanism that fits our model is truthful.*

*Proof.*

**Proof of IR** Let the notation of $\tilde{u}_i(t)$ and $u_i(t_i; t_i'; t_{-i})$ follows the definitions in Appendix D.1. Besides, denote $x^*(t) \in \arg\max_{x \in \mathcal{X}} \text{ASW}(x; t)$, $t_i^*(t_{-i}) \in \arg\min_{t_i \in \mathcal{T}_i} \max_{x \in \mathcal{X}} \text{ASW}(x; t_i^*; t_{-i})$ and $p_i(t)$ as $p_i$ in above definitions (note that $x^{-i} = x^*(t_i^*, t_{-i})$ in above definitions), respectively. By little computation we derive that,

$$
\begin{aligned}
\tilde{u}_i(t) =& v_i(x_i^*(t); t_i) - p_i(t) \\
=& v_i(x_i^*(t); t_i) - \frac{1}{w_i} \left[ \text{ASW}(x^*(t_i^*(t_{-i}); t_{-i}); t_i^*(t_{-i}), t_{-i}) - \text{ASW}_{-i}(x^*(t); t) \right] \\
=& \frac{1}{w_i} \left[ \text{ASW}(x^*(t); t) - \text{ASW}(x^*(t_i^*(t_{-i}); t_{-i}); t_i^*(t_{-i}), t_{-i}) \right]
\end{aligned}
$$

Recall that $t_i^*(t_{-i})$ minimizes $\text{ASW}(x^*(\cdot; t_{-i}); \cdot, t_{-i})$ by definition. Notice that the first term and second term are the realizations of this function with input $t_i$ and $t_i^*(t_{-i})$, respectively. Therefore $\tilde{u}_i(t) \geq 0$, which guarantees IR.

**Proof of IC** We compute the $u_i(t_i; t_i', t_{-i})$:

$$
\begin{aligned}
u_i(t_i; t_i', t_{-i}) =& v_i(x^*(t_i', t_{-i}); t_i) - p_i(t_i', t_{-i}) \\
=& v_i(x^*(t_i', t_{-i}); t_i) \\
& - \frac{1}{w_i} \left[ \text{ASW}(x^*(t_i^*(t_{-i}); t_{-i}); t_i^*(t_{-i}), t_{-i}) - \text{ASW}_{-i}(x^*(t_i', t_{-i}); t_i', t_{-i}) \right]
\end{aligned}
$$

Notice that $\text{ASW}(x^*(t_i^*(t_{-i}); t_{-i}); t_i^*(t_{-i}), t_{-i})$ does not rely on $t_i'$, therefore, we abbreviate this term as $h_i(t_{-i})$. Also notice that $\text{ASW}_{-i}(x; t_i', t_{-i})$ does not rely on $t_i'$, then,

$$
\begin{aligned}
u_i(t_i; t_i', t_{-i}) =& v_i(x^*(t_i', t_{-i}); t_i) + \frac{1}{w_i} \text{ASW}_{-i}(x^*(t_i', t_{-i}); t_i', t_{-i}) - \frac{h_i(t_{-i})}{w_i} \\
=& v_i(x^*(t_i', t_{-i}); t_i) + \frac{1}{w_i} \text{ASW}_{-i}(x^*(t_i', t_{-i}); t_i, t_{-i}) - \frac{h_i(t_{-i})}{w_i} \\
=& \frac{1}{w_i} \text{ASW}(x^*(t_i', t_{-i}); t_i, t_{-i}) - \frac{h_i(t_{-i})}{w_i} \\
\leq& \frac{1}{w_i} \text{ASW}(x^*(t_i, t_{-i}); t_i, t_{-i}) - \frac{h_i(t_{-i})}{w_i} \\
=& v_i(x^*(t_i, t_{-i}); t_i) + \frac{1}{w_i} \text{ASW}_{-i}(x^*(t_i, t_{-i}); t_i, t_{-i}) - \frac{h_i(t_{-i})}{w_i} \\
=& u_i(t_i; t_i, t_{-i})
\end{aligned}
$$

The inequality is because $x^*(t)$ is the maximizer of $\text{ASW}(x; t)$.

Above all, we complete the proof.

$\square$

# D OMITTED PROOFS

## D.1 PROOF OF THEOREM 3.4

**Theorem 3.4.** *Following mechanism classes are equivalent:* [13]

- *The class $\mathcal{M}^{D,IC}$ of direct mechanisms $\mathcal{M}^d = (\boldsymbol{x}, \boldsymbol{p})$ with IC property,*

- *The class $\mathcal{M}^M$ of menu mechanisms $\mathcal{M}^m$, where $\mathcal{M}^m = \{\mathcal{M}_i^m\}_{i \in [n]}$ and $\mathcal{M}_i^m = \{\mathcal{X}_i^m, p_i^m\}$,*

- *The class $\mathcal{M}^{FM,p}$ of full-menu mechanisms $\mathcal{M}^f$, where $\mathcal{M}^f = \{\mathcal{M}_i^f\}_{i \in [n]}$ and $\mathcal{M}_i^f = \{\mathcal{X}_i, p_i^f\}$, satisfying **pricing rule decomposition**.*

*Proof.* We only need to prove that, for the mechanism in one class, there is a mechanism in another class such that they are equivalent.

$(3) \Rightarrow (2)$   Trivial, since a full-menu mechanism satisfying pricing rule decomposition must be a menu mechanism.

$(2) \Rightarrow (1)$   Let $\mathcal{M}^m \in \mathcal{M}^M$ be a menu mechanism, we tend to show that we can transform $\mathcal{M}^m$ into an equivalent direct mechanism $\mathcal{M}^d$ such that $\mathcal{M}^d$ is IC.

We fix player $i$, player $i$'s type $\boldsymbol{t}_i$ and other players' types $\boldsymbol{t}_{-i}$. Then we need to show that player $i$ has no incentive to deviate from $\boldsymbol{t}_i$ in $\mathcal{M}^d$.

Denote $p(\boldsymbol{x}_i; \boldsymbol{t}_{-i})$ as the pricing rule of $\mathcal{M}^m$ to player $i$. If player $i$ reports $\boldsymbol{t}_i$, she will get the allocation

$$\boldsymbol{x}_i^* \in \arg\max_{\boldsymbol{x}_i \in \mathcal{X}_i} v(\boldsymbol{x}_i; \boldsymbol{t}_i) - p(\boldsymbol{x}_i; \boldsymbol{t}_{-i})$$

and pay the price $p_i^* = p(\boldsymbol{x}_i^*; \boldsymbol{t}_{-i})$. Otherwise, if player $i$ reports $\boldsymbol{t}_i'$, denote that she gets the allocation $\boldsymbol{x}_i'$ and pays the price $p_i' = p(\boldsymbol{x}_i'; \boldsymbol{t}_{-i})$. We have

$$u_i(\boldsymbol{x}_i^*; p_i^*; \boldsymbol{t}_i) = v_i(\boldsymbol{x}_i^*; \boldsymbol{t}_i) - p_i^* \geq v_i(\boldsymbol{x}_i'; \boldsymbol{t}_i) - p_i' = u_i(\boldsymbol{x}_i'; p_i'; \boldsymbol{t}_i)$$

The inequality follows from that $\boldsymbol{x}_i^*$ maximizes $v(\boldsymbol{x}_i; \boldsymbol{t}_i) - p(\boldsymbol{x}_i; \boldsymbol{t}_{-i})$. Notice that LHS is the utility of truth-telling and RHS is the utility of deviation. Thus, the IC of $\mathcal{M}^d$ follows directly.

$(1) \Rightarrow (3)$   Let $\mathcal{M}^d = (\boldsymbol{x}^d, \boldsymbol{p}^d)$ be a mechanism that satisfies IC, we need to construct a full-menu mechanism $\mathcal{M}^f$ with pricing rule $\{p_i^f(\boldsymbol{x}_i; \boldsymbol{t}_{-i})\}_{i \in [n]}$ satisfying pricing rule decomposition, and show that $\mathcal{M}^f$ is equivalent with $\mathcal{M}^d$.

We introduce two notations here. Denote $u_i(\boldsymbol{t}_i'; \boldsymbol{t})$ is the utility of player $i$ in $\mathcal{M}^d$ when the reported type profile is $\boldsymbol{t}$ and her true type is $\boldsymbol{t}_i'$. Denote $\tilde{u}_i(\boldsymbol{t}) = u_i(\boldsymbol{t}_i; \boldsymbol{t})$ is the utility of player $i$ in $\mathcal{M}^d$ when she reports the true type.

We first shed light on how we construct $p_i^f(\boldsymbol{x}_i; \boldsymbol{t}_{-i})$. Notice that the IC condition of $\mathcal{M}^d$ shows that,

$$\langle \boldsymbol{t}_i, \boldsymbol{x}_i^d(\boldsymbol{t}_i'; \boldsymbol{t}_{-i}) \rangle + c_i(\boldsymbol{x}_i^d(\boldsymbol{t}_i'; \boldsymbol{t}_{-i})) - p_i^d(\boldsymbol{t}_i'; \boldsymbol{t}_{-i}) \leq \langle \boldsymbol{t}_i, \boldsymbol{x}_i^d(\boldsymbol{t}_i; \boldsymbol{t}_{-i}) \rangle + c_i(\boldsymbol{x}_i^d(\boldsymbol{t}_i; \boldsymbol{t}_{-i})) - p_i^d(\boldsymbol{t}_i; \boldsymbol{t}_{-i}) \tag{3}$$

If we take $\boldsymbol{x}_i^d(\boldsymbol{t}_i'; \boldsymbol{t}_{-i})$ and $p_i^d(\boldsymbol{t}_i'; \boldsymbol{t}_i)$ as free variables $\boldsymbol{x}_i$ and $p_i$, then Equation (3) becomes,

$$p_i \geq -\tilde{u}_i(\boldsymbol{t}) + c_i(\boldsymbol{x}_i) + \langle \boldsymbol{t}_i, \boldsymbol{x}_i \rangle \tag{4}$$

where $\tilde{u}_i(\boldsymbol{t}) = v_i(\boldsymbol{x}_i^d(\boldsymbol{t}); \boldsymbol{t}_i) - p_i^d(\boldsymbol{t}) = \langle \boldsymbol{t}_i, \boldsymbol{x}_i^d(\boldsymbol{t}_i; \boldsymbol{t}_{-i}) \rangle + c_i(\boldsymbol{x}_i^d(\boldsymbol{t}_i; \boldsymbol{t}_{-i})) - p_i^d(\boldsymbol{t}_i; \boldsymbol{t}_{-i})$ is constant *w.r.t.* $\boldsymbol{x}_i$ and $p_i$.

---

[13]Hammond (1979) derived the relation between IC mechanism and menu mechanism, while Rochet (1987) derived the convex utility function in truthful mechanism, we argue that our characterization results are different from theirs and in fact more general. See Appendix A for more details.

When $\boldsymbol{t}_i = \boldsymbol{t}_i'$, Equation (3) takes equality on two sides. We suspect that Equation (4) should also take the equality sometimes, therefore, $p_i$ should be the minimum value such that Equation (4) always hold.

Inspired on above, we define

$$
\begin{aligned}
p_i^f(\boldsymbol{x}_i; \boldsymbol{t}_{-i}) &= \sup_{\boldsymbol{t}_i \in \mathcal{T}_i} -\tilde{u}_i(\boldsymbol{t}) + c_i(\boldsymbol{x}_i) + \langle \boldsymbol{t}_i, \boldsymbol{x}_i \rangle \\
&= c_i(\boldsymbol{x}_i) + \sup_{\boldsymbol{t}_i \in \mathcal{T}_i} -\tilde{u}_i(\boldsymbol{t}) + \langle \boldsymbol{t}_i, \boldsymbol{x}_i \rangle
\end{aligned}
\tag{5}
$$

Next, we will show that $\{p_i^f(\boldsymbol{x}_i; \boldsymbol{t}_{-i})\}_{i \in [n]}$-represented mechanism $\mathcal{M}^f$ is equivalent to $(\boldsymbol{x}^d, \boldsymbol{p}^d)$-represented mechanism $\mathcal{M}^d$. To show this, we need to show following statements,

1. $p_i^f(\boldsymbol{x}_i; \boldsymbol{t}_{-i})$ satisfies *pricing rule decomposition*,

2. $\boldsymbol{x}_i^d(\boldsymbol{t}) \in \arg\max_{\boldsymbol{x}_i \in \mathcal{X}_i} v_i(\boldsymbol{x}_i; \boldsymbol{t}_i) - p_i^f(\boldsymbol{x}_i; \boldsymbol{t}_{-i})$,

3. $p_i^d(\boldsymbol{t}) = p_i^f(\boldsymbol{x}_i^d(\boldsymbol{t}); \boldsymbol{t}_{-i})$,

where the second condition states that the allocation of $\mathcal{M}^d$ equals the allocation of $\mathcal{M}^f$, and the third condition states that they also charge same price to players.

PROVE THE FIRST STATEMENT    Notice that

$$
p_i^f(\boldsymbol{x}_i; \boldsymbol{t}_{-i}) - c_i(\boldsymbol{x}_i) = \sup_{\boldsymbol{t}_i \in \mathcal{T}_i} -\tilde{u}_i(\boldsymbol{t}_i; \boldsymbol{t}_{-i}) + \langle \boldsymbol{t}_i, \boldsymbol{x}_i \rangle
$$

is the Fenchel conjugate of $-\tilde{u}_i(\boldsymbol{t}_i; \boldsymbol{t}_{-i})$ *w.r.t.* $\boldsymbol{t}_i$. Therefore, it's convex by nature of Fenchel conjugate. (Boyd & Vandenberghe, 2004)

PROVE THE THIRD STATEMENT    By definition,

$$
\begin{aligned}
p_i^f(\boldsymbol{x}_i^d(\boldsymbol{t}); \boldsymbol{t}_{-i}) &= \sup_{\boldsymbol{t}_i' \in \mathcal{T}_i} p_i^d(\boldsymbol{t}_i'; \boldsymbol{t}_{-i}) + v_i(\boldsymbol{x}_i^d(\boldsymbol{t}); \boldsymbol{t}_i') - v_i(\boldsymbol{x}_i^d(\boldsymbol{t}_i'; \boldsymbol{t}_{-i}); \boldsymbol{t}_i') \\
&\geq p_i^d(\boldsymbol{t}_i; \boldsymbol{t}_{-i}), \qquad \text{by letting } \boldsymbol{t}_i' = \boldsymbol{t}_i
\end{aligned}
\tag{6}
$$

In order to prove the other side, we first observe that, by IC,

$$
\begin{aligned}
\tilde{u}_i(\boldsymbol{t}) &\geq u_i(\boldsymbol{t}_i; \boldsymbol{t}_i', \boldsymbol{t}_{-i}) \\
\Leftrightarrow v_i(\boldsymbol{x}_i^d(\boldsymbol{t}); \boldsymbol{t}_i) - p_i^d(\boldsymbol{t}) &\geq v_i(\boldsymbol{x}_i^d(\boldsymbol{t}_i', \boldsymbol{t}_{-i}); \boldsymbol{t}_i) - p_i^d(\boldsymbol{t}_i', \boldsymbol{t}_{-i})
\end{aligned}
$$

By switching $\boldsymbol{t}_i$ and $\boldsymbol{t}_i'$ we get,

$$
\begin{aligned}
v_i(\boldsymbol{x}_i^d(\boldsymbol{t}_i', \boldsymbol{t}_{-i}); \boldsymbol{t}_i') - p_i^d(\boldsymbol{t}_i', \boldsymbol{t}_{-i}) &\geq v_i(\boldsymbol{x}_i^d(\boldsymbol{t}_i, \boldsymbol{t}_{-i}); \boldsymbol{t}_i') - p_i^d(\boldsymbol{t}_i, \boldsymbol{t}_{-i}) \\
\Leftrightarrow v_i(\boldsymbol{x}_i^d(\boldsymbol{t}_i, \boldsymbol{t}_{-i}); \boldsymbol{t}_i') - v_i(\boldsymbol{x}_i^d(\boldsymbol{t}_i', \boldsymbol{t}_{-i}); \boldsymbol{t}_i') &\leq p_i^d(\boldsymbol{t}_i, \boldsymbol{t}_{-i}) - p_i^d(\boldsymbol{t}_i', \boldsymbol{t}_{-i})
\end{aligned}
$$

Taking it into Equation (6), we derive,

$$
\begin{aligned}
p_i^f(\boldsymbol{x}_i^d(\boldsymbol{t}); \boldsymbol{t}_{-i}) &= \sup_{\boldsymbol{t}_i' \in \mathcal{T}_i} p_i^d(\boldsymbol{t}_i'; \boldsymbol{t}_{-i}) + v_i(\boldsymbol{x}_i^d(\boldsymbol{t}); \boldsymbol{t}_i') - v_i(\boldsymbol{x}_i^d(\boldsymbol{t}_i'; \boldsymbol{t}_{-i}); \boldsymbol{t}_i') \\
&\leq \sup_{\boldsymbol{t}_i' \in \mathcal{T}_i} p_i^d(\boldsymbol{t}_i'; \boldsymbol{t}_{-i}) + p_i^d(\boldsymbol{t}_i, \boldsymbol{t}_{-i}) - p_i^d(\boldsymbol{t}_i', \boldsymbol{t}_{-i}) \\
&= p_i^d(\boldsymbol{t}_i, \boldsymbol{t}_{-i})
\end{aligned}
\tag{7}
$$

Together with Equation (6) and Equation (7), we finish this part.

PROVE THE SECOND STATEMENT   We have

$$v_i(\boldsymbol{x}_i^d(\boldsymbol{t}); \boldsymbol{t}_i) - p_i^f(\boldsymbol{x}_i^d(\boldsymbol{t}); \boldsymbol{t}_{-i})$$
$$= v_i(\boldsymbol{x}_i^d(\boldsymbol{t}); \boldsymbol{t}_i) - p_i^d(\boldsymbol{t}) = \tilde{u}_i(\boldsymbol{t})$$

We need to prove $v_i(\boldsymbol{x}_i; \boldsymbol{t}_i) - p_i^f(\boldsymbol{x}_i; \boldsymbol{t}_{-i}) \le \tilde{u}_i(\boldsymbol{t})$ for all $\boldsymbol{x}_i \in \mathcal{X}_i$.

Notice that

$$\text{LHS} = v_i(\boldsymbol{x}_i; \boldsymbol{t}_i) - c_i(\boldsymbol{x}_i) - \left( \sup_{\boldsymbol{t}_i' \in \mathcal{T}_i} \tilde{u}_i(\boldsymbol{t}_i', \boldsymbol{t}_{-i}) + \langle \boldsymbol{t}_i', \boldsymbol{x}_i \rangle \right)$$
$$\le v_i(\boldsymbol{x}_i; \boldsymbol{t}_i) - c_i(\boldsymbol{x}_i) - \tilde{u}_i(\boldsymbol{t}) - \langle \boldsymbol{t}_i, \boldsymbol{x}_i \rangle, \qquad \text{let } \boldsymbol{t}_i' = \boldsymbol{t}_i$$
$$= \tilde{u}_i(\boldsymbol{t})$$

Hence we complete the proof.

$\square$

## D.2   PROOF OF THEOREM 3.5

**Theorem 3.5.** *Following mechanism classes are equivalent:*

- *The class $\mathcal{M}^{D,T}$ of truthful direct mechanisms $\mathcal{M}^d = (\boldsymbol{x}, \boldsymbol{p})$ (IC & IR),*

- *The class $\mathcal{M}^{FM,pn}$ of full-menu mechanisms $\mathcal{M}^f$, where $\mathcal{M}^f = \{\mathcal{M}_i^f\}_{i \in [n]}$ and $\mathcal{M}_i^f = \{\mathcal{X}_i, p_i^f\}$, satisfying **pricing rule decomposition** and **no-buy-no-pay**.*

*Proof.* The line of this proof follows similar with those in Theorem 3.4. We also let the notations follow those in proof of Theorem 3.4

$(2) \Rightarrow (1)$   Let $\mathcal{M}^f$ be a full-menu mechanism satisfying *pricing rule decomposition* and *no-buy-no-pay* and $\mathcal{M}^d$ be corresponding direct mechanism. By Theorem 3.4 we know that $\mathcal{M}^d$ is IC. We then show that $\mathcal{M}^d$ is also IR.

Notice that player $i$'s utility in $\mathcal{M}^d$ is

$$\max_{\boldsymbol{x}_i \in \mathcal{X}_i} v_i(\boldsymbol{x}_i; \boldsymbol{t}_i) - p_i^f(\boldsymbol{x}_i; \boldsymbol{t}_{-i})$$
$$\ge v_i(\boldsymbol{0}; \boldsymbol{t}_i) - p_i^f(\boldsymbol{0}; \boldsymbol{t}_{-i})$$
$$= - p_i^f(\boldsymbol{0}; \boldsymbol{t}_{-i}) \ge 0$$

$(1) \Rightarrow (2)$   Let $\mathcal{M}^d$ be a truthful direct mechanism and $p_i^f(\boldsymbol{x}_i; \boldsymbol{t}_{-i})$ be the pricing rule constructed in Appendix D.1. By Theorem 3.4 we already know that $p_i^f$-represented mechanism satisfies *pricing rule decomposition*. For *no-buy-no-pay*, we have

$$p_i^f(\boldsymbol{0}; \boldsymbol{t}_{-i}) = \sup_{\boldsymbol{t}_i \in \mathcal{T}_i} -\tilde{u}_i(\boldsymbol{t}) + c_i(\boldsymbol{0}) + \langle \boldsymbol{t}_i, \boldsymbol{0} \rangle$$
$$= \sup_{\boldsymbol{t}_i \in \mathcal{T}_i} -\tilde{u}_i(\boldsymbol{t}) \le 0$$

where the inequality comes from IR, which says that truthful telling gives non-negative utility, which is exactly $\tilde{u}_i(\boldsymbol{t})$.

Above all, we complete the proof.

$\square$

### D.3    PROOF OF PROPOSITION 5.1

**Proposition 5.1.** *The mechanism class $\mathcal{M}^{PFM}$ is a universal approximator for the mechanism class $\mathcal{M}^{FM,pn}$, if the pricing functions are represented by MoA, LSE, GroupMax, or GroupLSE.*

*Proof.*

$\mathcal{M}^{PFM} \subseteq \mathcal{M}^{FM,pn}$**:**    Whether functions are parameterized by MoA, LSE, GroupMax or GroupMSE, the function is convex on $\boldsymbol{x}_i$ and have no constraints on $\boldsymbol{t}_{-i}$ by nature of the structure of them. Besides, the *no-buy-no-pay* constraint satisfies by design. Therefore, $\mathcal{M}^{PFM} \subseteq \mathcal{M}^{FM,pn}$.

$\varepsilon > 0$ **approximation:**    Notice that GroupMax can express MoA, GroupLSE can express LSE and LSE can arbitrarily approximate MoA. We only need to consider the class of MoA.

Kim & Kim (2022) shows that parameterized max-of-affine functions are universal approximators of functions those are continuous, convex on some input $\boldsymbol{x}$ and have no constraints on other input $\boldsymbol{y}$.

Fix any $\varepsilon > 0$. Let $p_i(\boldsymbol{x}_i; \boldsymbol{t}_{-i})$ be such a convex function that $p_i(\mathbf{0}; \boldsymbol{t}_{-i}) \leq 0$ and $p_i(\boldsymbol{x}_i; \boldsymbol{t}_{-i}; \theta)$ be a parameterized function such that $l_\infty(p_i, p_i(\cdot; \cdot; \theta) \leq \frac{\varepsilon}{2}$.

We construct another function $q_i(\boldsymbol{x}_i; \boldsymbol{t}_{-i}; \theta) = p_i(\boldsymbol{x}_i; \boldsymbol{t}_{-i}; \theta) - \frac{\varepsilon}{2}$. Since $p_i(\boldsymbol{x}_i; \boldsymbol{t}_{-i}; \theta)$ is a realization of PMA and $q_i$ has only constant difference with $p_i$, thus $q_i$ is also a realization of PMA.

We have $l_\infty(p_i, q_i(\cdot; \cdot; \theta)) \leq l_\infty(p_i(\cdot; \cdot; \theta), q_i(\cdot; \cdot; \theta)) + l_\infty(p_i, p_i(\cdot; \cdot; \theta)) \leq \frac{\varepsilon}{2} + \frac{\varepsilon}{2} = \varepsilon$ and $q_i(\mathbf{0}; \boldsymbol{t}_{-i}; \theta) = p_i(\mathbf{0}; \boldsymbol{t}_{-i}; \theta) - \frac{\varepsilon}{2} \leq p_i(\mathbf{0}; \boldsymbol{t}_{-i}) + \frac{\varepsilon}{2} - \frac{\varepsilon}{2} = p_i(\mathbf{0}; \boldsymbol{t}_{-i}) \leq 0$, that completes the proof.

$\square$

### D.4    PROOF OF THEOREM 5.4

**Theorem 5.4.** *Assume that $\mathcal{M}^1$ is a universal approximator of $\mathcal{M}$ under following technical conditions,*

*1. $v_0(x; t)$ is continuous on $\mathcal{X}$ and $\mathcal{T}$ (thus continuous consistently).*

*2. The pricing function $p(x, t)$ is $\varepsilon_1$-strongly convex on $x$ for some $\varepsilon_1 > 0$, when $p \in \mathcal{M}$.*

*Then, $\mathrm{MEU}(\mathcal{M}^1) = \mathrm{MEU}(\mathcal{M})$.*

*Proof.* We assume $\lambda = 1$ without loss of generality. We denote $u(p)$ the objective function of $p \in \mathcal{M}$ in Equation (1). We only need to prove that for any $\varepsilon > 0$ and any $p \in \mathcal{M}$, there is $p_1 \in \mathcal{M}^1$ such that $u(p_1) > u(p) - \varepsilon$. To do this, we first derive a lemma demonstrating the "continuity" property of $x(t)$ over $l_\infty$ of $p(x, t)$.

**Lemma D.1.** *Let $p_1(x, t), p_2(x, t)$ be two pricing functions such that $l_\infty(p_1, p_2) \leq \varepsilon$ and $p_1$ is $\varepsilon_1$-convex on $x$, denote $x_1^*(t) = \arg \max_{x \in \mathcal{X}} \langle x, t \rangle - p_1(x, t)$ and $x_2^*(t) = \arg \max_{x \in \mathcal{X}} \langle x, t \rangle - p_2(x, t)$, then, we have that*

$$\|x_1^*(t) - x_2^*(t)\| \leq 2\sqrt{\frac{\varepsilon}{\varepsilon_1}}$$

*proof of Lemma D.1.* Fix some $t \in \mathcal{T}$, by strong concavity we have that for all $x \in \mathcal{X}$ such that $\|x_1^*(t) - x\|_2 > \delta$ with $\delta = 2\sqrt{\frac{\varepsilon}{\varepsilon_1}}$, we have that $p_1(x, t) - p_1(x_1^*(t)) > \frac{\varepsilon_1 \delta^2}{2}$. Then,

$$
\begin{aligned}
&p_2(x, t) - p_2(x_1^*(t)) \\
=& p_2(x, t) - p_1(x, t) + p_1(x, t) - p_1(x_1^*(t)) + p_1(x_1^*(t)) - p_2(x_1^*(t)) \\
>& -2\varepsilon + \frac{\varepsilon_1 \delta^2}{2} \\
\geq& 0
\end{aligned}
$$

It shows that such $x$ cannot be the maximum point of $p_2(x, t)$. Therefore, we must have $\|x_2^*(t) - x_1^*(t)\|_2 \leq \delta = 2\sqrt{\frac{\varepsilon}{\varepsilon_1}}$, which completes the proof. $\qquad\square$

Now we continue the original proof. We also need an observation that, by optimality of $x_1(t)$,

$$\langle x_1(t), t \rangle - p_1(x_1(t), t) \geq \langle x_2(t), t \rangle - p_1(x_2(t), t)$$
$$p_1(x_2(t); t) \geq p_1(x_1(t), t) + \langle x_2(t) - x_1(t), t \rangle$$

By consistent continuity of $v_0(x; t)$, we know that there exists $\delta_1 > 0$ such that $\|x_1 - x_2\| \leq \delta_1$ indicates that $v_0(x_1, t) - v_0(x_2, t) \leq \frac{\varepsilon}{2}$. Denote $\delta_2 = \min\{\delta_1, \frac{\varepsilon}{4T}\}$, where $T = \max_{t \in \mathcal{T}} \|t\|_2$. We let $\delta_3 = \frac{\delta_2^2 \varepsilon_1}{4} > 0$ such that as long as $l_\infty(p_1, p_2) \leq \delta_3$ holds and $p_1$ is $\varepsilon_1$-strong convex, we have $\|x_2(t) - x_1(t)\| \leq \delta_2$, Take $\delta = \min\{\frac{\varepsilon}{4}, \delta_3\}$, while $l_\infty(p_1, p_2) \leq \delta$ holds, we have that

$$\begin{aligned}
p_2(x_2(t), t) &\geq p_1(x_2(t), t) - \frac{\varepsilon}{4} \\
&\geq p_1(x_1(t); t) + \langle x_2(t) - x_1(t), t \rangle - \frac{\varepsilon}{4} \\
&\geq p_1(x_1(t); t) - T\|x_2(t) - x_1(t)\| - \frac{\varepsilon}{4} \\
&\geq p_1(x_1(t); t) - \frac{\varepsilon}{4} - \frac{\varepsilon}{4} \\
&\quad \cdots \text{because } l_\infty(p_1, p_2) \leq \delta_3 \text{ and then } \|x_2(t) - x_1(t)\|_2 \leq \delta_2 \leq \frac{\varepsilon}{4T} \\
&= p_1(x_1(t); t) - \frac{\varepsilon}{2}.
\end{aligned}$$

Also note that $\|x_2(t) - x_1(t)\|_2 \leq \delta_2 \leq \delta_1$, thus $v_0(x_1(t), t) - v_0(x_2(t), t) \leq \frac{\varepsilon}{2}$. Summing up the arguments above, we have that

$$v_0(x_2(t), t) + p_2(x_2(t), t) \geq v_0(x_1(t), t) + p_1(x_1(t), t) - \varepsilon.$$

This concludes the proof.

$\qquad\square$

### D.5 PROOF OF PROPOSITION 5.5

**Proposition 5.5.** *Consider an AMA mechanism $M^{AMA}$ with positive weights $w_1, \ldots, w_n$ and a shift function $\lambda(\boldsymbol{x})$. Assume more that an oracle $\mathcal{O}^{AMA}$ of AMA mechanism exists that can run the mechanism $M^{AMA}$ under input $\boldsymbol{t}$. Formally, $\mathcal{O}^{AMA}$ receives $M^{AMA}$ (or equivalently, $\boldsymbol{w}$ and $\lambda$) and $\boldsymbol{t}$ as inputs, and output the resulting allocation $\boldsymbol{x}$ and payment $\boldsymbol{p}$.*

*Given any AMA mechanism $M^{AMA}$, we can explicitly construct a full-menu mechanism $M^F$ with pricing functions $\{p_i^f(\boldsymbol{x}_i; \boldsymbol{t}_{-i})\}_{i \in [n]}$, that receives type profile $\boldsymbol{t}$, outputs the full menu $p_i : \mathcal{X}_i \to \mathbb{R}, i \in [n]$, and is equivalent to $M^{AMA}$.*

*Additionally, querying $\{p_i(\boldsymbol{x}_i)\}_{i \in [n]}$ at some point $\boldsymbol{x} \in \mathcal{X}$ needs polynomial-time computation and $O(n)$ black-box queries of the oracle $\mathcal{O}^{AMA}$.*

We have extended the AMA mechanism to our model and show that extended AMA is truthful in Appendix B and Appendix C.

*Proof of Proposition 5.5.* Denote $p^f$ as the pricing rule of the full menu. We construct $p_i^f$ as follows given $\boldsymbol{x}_i$ and $\boldsymbol{t}_{-i}$,

$$\boldsymbol{t}_i^*(\boldsymbol{t}_{-i}) \in \underset{\boldsymbol{t}_i \in \mathcal{T}}{\arg\min} \max_{\boldsymbol{x} \in \mathcal{X}} \mathrm{ASW}(\boldsymbol{x}; \boldsymbol{t}_i; \boldsymbol{t}_{-i})$$

$$\boldsymbol{x}^{-*}(\boldsymbol{t}_{-i}) \in \underset{\boldsymbol{x} \in \mathcal{X}}{\arg\max} \, \mathrm{ASW}(\boldsymbol{x}; \boldsymbol{t}_i^*(\boldsymbol{t}_{-i}), \boldsymbol{t}_{-i})$$

$$\boldsymbol{x}_{-i}^{i,*}(\boldsymbol{x}_i, \boldsymbol{t}_{-i}) \in \underset{\boldsymbol{x}_{-i} \in \mathcal{X}}{\arg\max} \, \mathrm{ASW}(\boldsymbol{x}_i, \boldsymbol{x}_{-i}; \boldsymbol{t}) \quad \cdots \text{notice that the optimal } \boldsymbol{x}_{-i} \text{ do not depend on } \boldsymbol{t}_i$$

$$= \underset{\boldsymbol{x}_{-i} \in \mathcal{X}}{\arg\max} \, \mathrm{ASW}_{-i}(\boldsymbol{x}_i, \boldsymbol{x}_{-i}; \boldsymbol{t}_{-i}) + v_i(\boldsymbol{x}_i; \boldsymbol{t}_i)$$

$$= \underset{\boldsymbol{x}_{-i} \in \mathcal{X}}{\arg\max} \, \mathrm{ASW}_{-i}(\boldsymbol{x}_i, \boldsymbol{x}_{-i}; \boldsymbol{t}_{-i})$$

$$p_i^f(\boldsymbol{x}_i; \boldsymbol{t}_{-i}) = \frac{1}{w_i} \left[ \mathrm{ASW}(\boldsymbol{x}^{-*}(\boldsymbol{t}_{-i}); \boldsymbol{t}_i^*(\boldsymbol{t}_{-i}), \boldsymbol{t}_{-i}) - \mathrm{ASW}_{-i}(\boldsymbol{x}_i, \boldsymbol{x}_{-i}^{i,*}(\boldsymbol{x}_i, \boldsymbol{t}_{-i}); \boldsymbol{t}_{-i}) \right]$$

**Proof of equivalence to AMA** Next we show that such mechanism is equivalent to AMA. We begin with the utility of player $i$ with type $\boldsymbol{t}_i$ buying $\boldsymbol{x}_i$:

$$u_i(\boldsymbol{x}_i; \boldsymbol{t}) = v_i(\boldsymbol{x}_i; \boldsymbol{t}_i) - p_i^f(\boldsymbol{x}_i; \boldsymbol{t}_{-i})$$

$$= v_i(\boldsymbol{x}_i; \boldsymbol{t}_i) - \frac{1}{w_i} \left[ \mathrm{ASW}(\boldsymbol{x}^{-*}; \boldsymbol{t}_i^*, \boldsymbol{t}_{-i}) - \mathrm{ASW}_{-i}(\boldsymbol{x}_i, \boldsymbol{x}_{-i}^{i,*}(\boldsymbol{x}_i, \boldsymbol{t}_{-i}); \boldsymbol{t}_{-i}) \right]$$

$$= \mathrm{ASW}(\boldsymbol{x}_i, \boldsymbol{x}_{-i}^{i,*}(\boldsymbol{x}_i, \boldsymbol{t}_{-i}); \boldsymbol{t}) - \frac{h_i(\boldsymbol{t}_{-i})}{w_i}$$

$$\leq \mathrm{ASW}(\boldsymbol{x}^*(\boldsymbol{t}); \boldsymbol{t}) - \frac{h_i(\boldsymbol{t}_{-i})}{w_i}$$

The inequality follows from that $\boldsymbol{x}^*$ is the maximizer. When player $i$ choose to buy $\boldsymbol{x}_i^*(\boldsymbol{t})$, we know that,

$$\boldsymbol{x}_{-i}^*(\boldsymbol{t}) = \boldsymbol{x}_{-i}^{i,*}(\boldsymbol{x}_i^*(\boldsymbol{t}), \boldsymbol{t}_{-i})$$

because $\boldsymbol{x}_{-i}^*(\boldsymbol{t})$ makes $\mathrm{ASW}(\boldsymbol{x}_i^*(\boldsymbol{t}), \boldsymbol{x}_{-i}; \boldsymbol{t})$ get its maximum *w.r.t.* $\boldsymbol{x}_{-i}$. Then, the utility of player $i$ equals to $\mathrm{ASW}(\boldsymbol{x}^*(\boldsymbol{t}); \boldsymbol{t}) - \frac{h_i(\boldsymbol{t}_{-i})}{w_i}$. It means that utility-maximizing players will definitely choose $\boldsymbol{x}^*(\boldsymbol{t})$. The equivalence of price is obvious based on this, thus we complete the proof of equivalence.

$O(n)$ **queries of** $\mathcal{O}^{AMA}$ Notice that the oracle $\mathcal{O}^{AMA}$ is a black box and we can only have access to the output allocation and price.

Now we focus on computing the price $p_i^f(\boldsymbol{x}_i; \boldsymbol{t}_{-i})$. The first term is $\mathrm{ASW}(\boldsymbol{x}^{-*}(\boldsymbol{t}_{-i}); \boldsymbol{t}_i^*(\boldsymbol{t}_{-i}), \boldsymbol{t}_{-i})$. Since this term has no relation with $\boldsymbol{t}_i$ or $\boldsymbol{x}_i$, thus it can be easily derived from AMA. Actually, by nature of AMA (in Step 4 in Definition B.5), we know that

$$\mathrm{ASW}(\boldsymbol{x}^{-*}(\boldsymbol{t}_{-i}); \boldsymbol{t}_i^*(\boldsymbol{t}_{-i}), \boldsymbol{t}_{-i}) = w_i \cdot p_i^{AMA}(\boldsymbol{t}) + \mathrm{ASW}_{-i}(\boldsymbol{x}^{AMA}(\boldsymbol{t}); \boldsymbol{t})$$

where $x^{AMA}$ and $p^{AMA}$ is the AMA allocation and payment rule, thus can be achieve from $\mathcal{O}^{AMA}$.

A more tricky one is to compute the second term $\mathrm{ASW}_{-i}(\boldsymbol{x}_i, \boldsymbol{x}_{-i}^{i,*}(\boldsymbol{x}_i, \boldsymbol{t}_{-i}); \boldsymbol{t}_{-i})$. We construct another society with $n - 1$ players, except player $i$, and let $\lambda^i(\boldsymbol{x}_{-i}) := \lambda(\boldsymbol{x}_i, \boldsymbol{x}_{-i})$. Then $\boldsymbol{x}_{-i}^{i,*}(\boldsymbol{x}_i, \boldsymbol{t}_{-i})$ is the allocation in the AMA mechanism with weights $\boldsymbol{w}_{-i}$ and shift $\lambda^i(\boldsymbol{x}_{-i})$. We can call $\mathcal{O}^{AMA}$ with $(\boldsymbol{w}_{-i}, \lambda^i, \boldsymbol{t}_{-i})$ to get the output of $\boldsymbol{x}_{-i}^{i,*}(\boldsymbol{x}_i, \boldsymbol{t}_{-i})$, and then computing $\mathrm{ASW}_{-i}(\boldsymbol{x}_i, \boldsymbol{x}_{-i}^{i,*}(\boldsymbol{x}_i, \boldsymbol{t}_{-i}); \boldsymbol{t}_{-i})$.

Above all, computing the price in given $\boldsymbol{x}$ needs $n + 1 = O(n)$ query of $\mathcal{O}^{AMA}$. The other computation lies in computing affine social welfare, which can be directed computed in polynomial time.

$\square$

# E    DETAILS ABOUT LEARNING ALGORITHMS

## E.1    DERIVATION OF LEARNING ALGORITHMS

In this part, we derive the learning procedure to this problem.

To begin with, we present the optimization problem as follows,

$$\max_{\substack{\theta \in \Theta \\ \boldsymbol{x}_i^*(\boldsymbol{t}_i; \boldsymbol{t}_{-i}; \theta), i \in [n]}} \mathbb{E}_{\boldsymbol{t} \sim \mathcal{F}}[u_0(\boldsymbol{x}^*(\boldsymbol{t}; \theta), \boldsymbol{p}(\boldsymbol{x}^*(\boldsymbol{t}; \theta); \boldsymbol{t}; \theta); \boldsymbol{t})]$$

$$\text{s.t.}\quad \boldsymbol{x}_i^*(\boldsymbol{t}_i; \boldsymbol{t}_{-i}; \theta) \in \arg\max_{\boldsymbol{x}_i \in \mathcal{X}_i} u_i(\boldsymbol{x}_i, p_i(\boldsymbol{x}_i; \boldsymbol{t}_{-i}; \theta); \boldsymbol{t}_i) \qquad \forall \boldsymbol{t} \in \mathcal{T}, \forall i \in [n] \tag{8}$$

The algorithm control the pricing rule (represented by $\theta$) as well as the simulated players' utility-maximizing behaviors $\boldsymbol{x}_i^*(\cdot; \boldsymbol{t}_{-i}; \theta)$ for all $i$. $\boldsymbol{x}^*(\boldsymbol{t}; \theta)$ is short for $\{\boldsymbol{x}_i^*(\boldsymbol{t}_i; \boldsymbol{t}_{-i}; \theta)\}_{i \in [n]}$ and $\boldsymbol{p}(\boldsymbol{x}; \boldsymbol{t}; \theta)$ is short for $\{p_i(\boldsymbol{x}_i; \boldsymbol{t}_{-i}; \theta)\}_{i \in [n]}$

The **first step** is to sample $B$ size of i.i.d. samples from distribution $\mathcal{F}$. [14] We denote $\boldsymbol{t}^k$ as the $k$'th sample, $\mathcal{T}^B = \{\boldsymbol{t}^k\}_{1 \leq k \leq B}$ as the set of samples and $U(\mathcal{T}^B)$ as the uniform distribution on these samples. We then optimize the experience expected utility:

$$\max_{\substack{\theta \in \Theta \\ \boldsymbol{x}_i^*(\boldsymbol{t}_i; \boldsymbol{t}_{-i}; \theta), i \in [n]}} \mathbb{E}_{\boldsymbol{t} \sim \mathcal{U}(\mathcal{T}^B)}[u_0(\boldsymbol{x}^*(\boldsymbol{t}; \theta), \boldsymbol{p}(\boldsymbol{x}^*(\boldsymbol{t}; \theta); \boldsymbol{t}; \theta); \boldsymbol{t})]$$

$$\text{s.t.}\quad \boldsymbol{x}_i^*(\boldsymbol{t}_i; \boldsymbol{t}_{-i}; \theta) \in \arg\max_{\boldsymbol{x}_i \in \mathcal{X}_i} u_i(\boldsymbol{x}_i, p_i(\boldsymbol{x}_i; \boldsymbol{t}_{-i}; \theta); \boldsymbol{t}_i) \qquad \forall \boldsymbol{t} \in \mathcal{T}^B, \forall i \in [n] \tag{9}$$

Since there are only finite values of $\boldsymbol{t}$ in $\mathcal{T}^B$, we use $\{\boldsymbol{x}_i^k\}_{k \in [B]}$ to represent $\boldsymbol{x}_i^*(\boldsymbol{t}_i^k; \boldsymbol{t}_{-i}^k; \theta)$. Then, the problem becomes,

$$\max_{\substack{\theta \in \Theta \\ \{\boldsymbol{x}^k \in \mathcal{X}\}_{k \in [B]}}} \frac{1}{B} \sum_{k=1}^{B} \left[ u_0(\boldsymbol{x}^k, \boldsymbol{p}(\boldsymbol{x}^k; \boldsymbol{t}^k; \theta); \boldsymbol{t}^k) \right]$$

$$\text{s.t.}\quad \boldsymbol{x}_i^k \in \arg\max_{\boldsymbol{x}_i \in \mathcal{X}_i} u_i(\boldsymbol{x}_i, p_i(\boldsymbol{x}_i; \boldsymbol{t}_{-i}^k; \theta); \boldsymbol{t}_i^k) \qquad \forall k \in [B], \forall i \in [n] \tag{10}$$

Notice that the constraint we need to tackle is *function maximizer constraint*, to resolve this constraint, the **second step** is to utilize the method of envelope theorem (Milgrom & Segal, 2002).

To show how envelope theorem works, we first rewrite the maximizer constraint into *equality constraint* as follows,

$$u_i(\boldsymbol{x}_i^k, p_i(\boldsymbol{x}_i^k; \boldsymbol{t}_{-i}^k; \theta); \boldsymbol{t}_i^k) = \max_{\boldsymbol{x}_i \in \mathcal{X}_i} u_i(\boldsymbol{x}_i, p_i(\boldsymbol{x}_i; \boldsymbol{t}_{-i}^k; \theta); \boldsymbol{t}_i^k) \qquad \forall k \in [B], \forall i \in [n]$$

We denote $\boldsymbol{x}_i^{k*}(\theta)$ as the maximizer of the right-hand side (RHS). To address any violation of this equality, we introduce a ReLU penalty function, with the penalty intensity controlled by a hyper-parameter $\lambda > 0$. Consequently, the problem formulation becomes:

$$\max_{\substack{\theta \in \Theta \\ \{\boldsymbol{x}^k \in \mathcal{X}\}_{k \in [B]}}} \text{OBJ}(\theta, \{\boldsymbol{x}^k\}; \lambda) = \frac{1}{B} \sum_{k=1}^{B} \left[ u_0(\boldsymbol{x}^k, \boldsymbol{p}(\boldsymbol{x}^k; \boldsymbol{t}^k; \theta); \boldsymbol{t}^k) \right]$$

$$- \lambda \cdot \frac{1}{B} \sum_{k=1}^{B} \sum_{i=1}^{n} \text{ReLU}\left( u_i\left(\boldsymbol{x}_i^{k*}(\theta), p_i(\boldsymbol{x}_i^{k*}(\theta); \boldsymbol{t}_{-i}^k; \theta); \boldsymbol{t}_i^k\right) - u_i\left(\boldsymbol{x}_i^k, p_i(\boldsymbol{x}_i^k; \boldsymbol{t}_{-i}^k; \theta); \boldsymbol{t}_i^k\right) \right) \tag{11}$$

---

[14] This step can be done because we assume an oracle that can sample arbitrary size i.i.d. samples.

As is commonly done in learning-based algorithms (Amari, 1993; Bottou, 2010), we only need to compute the first-order derivatives with respect to $\theta$ and $\{\boldsymbol{x}^k\}$ to optimize $\text{OBJ}(\theta, \{\boldsymbol{x}^k\})$. While the derivative with respect to $\boldsymbol{x}^k$ is straightforward, the derivative with respect to $\theta$ is more challenging because $\boldsymbol{x}_i^{k*}(\theta)$ depends on $\theta$. The most significant challenge is that the function $\boldsymbol{x}_i^{k*}(\theta)$ is unknown; even if we can obtain $\boldsymbol{x}_i^{k*}(\theta)$ for a specific $\theta$ through optimization, computing $\frac{\partial \boldsymbol{x}_i^{k*}}{\partial \theta}(\theta)$ seems to be infeasible.

However, according to the envelope theorem (Milgrom & Segal, 2002), when computing $\frac{\partial u_i}{\partial \theta}(\boldsymbol{x}_i^{k*}(\theta), p_i(\boldsymbol{x}_i^{k*}(\theta); \boldsymbol{t}_{-i}^k; \theta); \boldsymbol{t}_i^k)$, we can treat $\boldsymbol{x}_i^{k*}(\theta)$ as a constant. In other words,

$$\frac{\partial u_i}{\partial \theta}(\boldsymbol{x}_i^{k*}(\theta), p_i(\boldsymbol{x}_i^{k*}(\theta); \boldsymbol{t}_{-i}^k; \theta); \boldsymbol{t}_i^k) = \frac{\partial u_i}{\partial \theta}(\boldsymbol{x}_i^{k*}, p_i(\boldsymbol{x}_i^{k*}; \boldsymbol{t}_{-i}^k; \theta); \boldsymbol{t}_i^k)|_{\boldsymbol{x}_i^{k*} = \boldsymbol{x}_i^{k*}(\theta)}$$

For completeness, an insightful proof of a simplified version of the envelope theorem is provided in Appendix C.

Building on this, it suffices to obtain a good estimate of $\boldsymbol{x}_i^{k*}(\theta)$ in this algorithm.

To achieve this, the **third step** is to define $\boldsymbol{x}_i^{k*}$ as an approximation of $\boldsymbol{x}_i^{k*}(\theta)$, which we can optimize through the optimization procedure as follows.

$$\max_{\{\boldsymbol{x}^{k*} \in \mathcal{X}\}_{k \in [B]}} \text{OBJ}^*(\boldsymbol{x}^{k*}) = \frac{1}{B} \sum_{k=1}^{B} \sum_{i=1}^{n} \left[ u_i(\boldsymbol{x}_i^{k*}, p_i(\boldsymbol{x}_i^{k*}; \boldsymbol{t}_{-i}^k; \theta); \boldsymbol{t}_i^k) \right]$$

For a specific instance of $(\theta, \{\boldsymbol{x}^k\}_{k \in [B]}, \{\boldsymbol{x}^{k*}\}_{k \in [B]})$, if the two optimization problems achieve their optima simultaneously (when we consider the optimization problem w.r.t. some variables, fix the other variables constant), then $\theta$ is guaranteed to be the optimal mechanism representation in equation Equation (11). Moreover, if $\boldsymbol{x} = \boldsymbol{x}^*$, then $\theta$ is the optimal mechanism representation in original problem Equation (9).

Building on this, we can see that the problem is analogous to finding the equilibrium of a multiple-agents Stackelberg game (Von Stackelberg, 2010). In this game, the principle first chooses $\theta$, representing the platform's penalized expected utility. After seeing $\theta$, agents side will selects $\boldsymbol{x}^*$ to optimize $\text{OBJ}^*(\boldsymbol{x}^*; \theta)$, which corresponds to the players' expected utility. The principle's utility is then designated by $\text{OBJ}(\theta, \{\boldsymbol{x}^*\})$

In our algorithm, we optimize these two objective functions concurrently. A simple illustration of our algorithm has been provided in Figure 1 in the main body.

After the training process, we left $\boldsymbol{x}$ and $\boldsymbol{x}^*$ behind, only denote $\theta^*$ as the learned mechanism representation.

### E.2 Pseudo-codes

In this section, we present the pseudo-codes of our training and inference procedure.

---

**Algorithm 1:** Training procedure

---

**Input:** number of players and items $(n, m)$, the oracle for i.i.d. samples $\mathcal{O}$
**Output:** mechanism parameters $\theta$

1 **Define hyper-parameters:** sample size $B$, batch size $B_0$, mechanism iteration $T_0$, platform
   allocation iteration $T_1$, player allocation iteration $T_2$, epoch $T$
2 Sample $B$ i.i.d. samples $\boldsymbol{t}^1, ..., \boldsymbol{t}^B$ from distribution $\mathcal{F}$ with oracle $\mathcal{O}$
3 Initialize mechanism parameters $\theta$, platform allocation $\boldsymbol{x} = \{\boldsymbol{x}_i^k\}_{1 \leq k \leq B, 1 \leq i \leq n}$, player
   allocation $\boldsymbol{x}^* = \{\boldsymbol{x}_i^{k,*}\}_{1 \leq k \leq B, 1 \leq i \leq n}$, penalty intensity $\lambda$
4 **for** $t = 1, ..., T$ **do**
5 $\quad$ *Optimizing platform's objective*:
6 $\quad$ **for** $t_0 = 1, ..., T_0$ **do**
7 $\quad\quad$ Randomly sample $B_0$ batch of data on the sample points $\{(\boldsymbol{t}^k, \boldsymbol{x}^k, \boldsymbol{x}^{k,*})\}_{k \in B}$
8 $\quad\quad$ Fix $\boldsymbol{x}, \boldsymbol{x}^*$, compute $\mathrm{OBJ}(\boldsymbol{x}, \theta; \boldsymbol{x}^*)$, using $B_0$ samples of data
9 $\quad\quad$ Optimize $\theta$ through gradient of $\mathrm{OBJ}(\boldsymbol{x}, \theta; \boldsymbol{x}^*)$ for one iteration
10 $\quad$ **end**
11 $\quad$ **for** $t_1 = 1, ..., T_1$ **do**
12 $\quad\quad$ Fix $\theta, \boldsymbol{x}^*$, compute $\mathrm{OBJ}(\boldsymbol{x}, \theta; \boldsymbol{x}^*)$ on all samples
13 $\quad\quad$ Optimize $\boldsymbol{x}$ through gradient of $\mathrm{OBJ}(\boldsymbol{x}, \theta; \boldsymbol{x}^*)$ for one iteration
14 $\quad$ **end**
15
16 $\quad$ *Optimizing player's objective*:
17 $\quad$ **for** $t_2 = 1, ..., T_2$ **do**
18 $\quad\quad$ Fix $\theta$, compute $\mathrm{OBJ}^*(\boldsymbol{x}^*; \theta)$ on all samples
19 $\quad\quad$ Optimize $\boldsymbol{x}^*$ through gradient of $\mathrm{OBJ}^*(\boldsymbol{x}^*; \theta)$ for one iteration
20 $\quad$ **end**
21
22 $\quad$ increase $\lambda$ moderately
23 **end**
24 **return** mechanism parameters $\theta$.

---

**Algorithm 2:** Inference procedure

---

**Input:** mechanism $\theta$, a type profile of players $\boldsymbol{t}$
**Output:** the allocation $\boldsymbol{x} \in \mathcal{X}$ and price $\boldsymbol{p} \in \mathbb{R}^n$ on the type profile

1 **Define hyper-parameters:** the iteration time $T$ for optimizing allocation
2 Initialize allocation $\boldsymbol{x}$. **for** $t = 1, ..., T$ **do**
3 $\quad$ *Optimizing players' utility:*
4 $\quad$ Compute the players' utilities over $\boldsymbol{x}$, $\mathrm{OBJ}(\boldsymbol{x}; \theta)$
5 $\quad$ Optimize $\boldsymbol{x}$ through gradient of $\mathrm{OBJ}(\boldsymbol{x}; \theta)$ for one iteration
6 **end**
7 compute players' payments: $p_i = p(\boldsymbol{x}_i; \boldsymbol{t}_{-i}; \theta), i \in [n]$ **return** players' allocations $\boldsymbol{x}$, players'
   payments $\boldsymbol{p}$

---

# F    DISCUSSIONS

## F.1    DISCUSSIONS ON MODEL EXPRESSIVENESS

One deficiency of the model is that, the model expressiveness is limited since we assume $\mathcal{X} = \times_{i \in [n]} \mathcal{X}_i$, but it is not always the case. As an example, in the traditional auction model, the auctioneers' allocation space is $\mathcal{X} = \Delta_n$, assuming there is $n - 1$ bidders, since items can not be over-allocated. However, $\Delta_n$ can not be written as Cartesian product $\times_{i \in [n]} \mathcal{X}_i$. We call such constraint as platform's hard allocation constraint.

Our argument is following: such hard constraint can be model into the platform's valuation. As long as $\mathcal{X}$ is convex (this is satisfied in auction problem), we can rewrite the platform's valuation as

follows,

$$\hat{v}_0(\boldsymbol{x};\boldsymbol{t}) = \begin{cases} v_0(\boldsymbol{x};\boldsymbol{t}) & \text{if } \boldsymbol{x} \in \mathcal{X} \\ -\infty & \text{if } \boldsymbol{x} \notin \mathcal{X} \end{cases} \tag{12}$$

We can verify that such valuation $\hat{v}_0(\boldsymbol{x};\boldsymbol{t})$ is still convex. Although such model does not capture the allocation constraint directly, we know that an optimal mechanism will never choose the allocation $\boldsymbol{x} \notin \mathcal{X}$. Therefore, as long as we achieve the optimal mechanism in this model, we immediately achieve the optimal mechanism of the original problem with platform's hard constraint $\mathcal{X}$.

It naturally leads another question is that, such $\hat{v}_0$ is not continuous thus hard to optimize. Our next argument is that, we can make a continuous approximation to $\hat{v}_0(\boldsymbol{x};\boldsymbol{t})$, which makes the optimization easier. Specifically, we let

$$\tilde{v}_0(\boldsymbol{x};\boldsymbol{t}) = \begin{cases} v_0(\boldsymbol{x};\boldsymbol{t}) & \text{if } \boldsymbol{x} \in \mathcal{X} \\ v_0(\text{proj}(\boldsymbol{x},\mathcal{X});\boldsymbol{t}) - M \cdot \text{proj}(\boldsymbol{x},\mathcal{X}) & \text{if } \boldsymbol{x} \notin \mathcal{X} \end{cases} \tag{13}$$

As long as $v_0$ is L-Lipschitz (this is again satisfied in auction problem), choose $M \geq L$ will make $v_0$ concave, continuous and have full domain $\times_{i \in [n]} \mathcal{X}_i$. As long as $M$ is large enough, the optimal mechanism in Equation (13) can be arbitrary close to the optimal mechanism in Equation (12), therefore approximate the optimal mechanism of original problem.

However, it's not known whether the optimal mechanism of Equation (13) equals the optimal mechanism of Equation (12) in general, for some constant of $M$. If this statement is true, we believe that such generalized model with flexible platform valuations can be seen as an equivalent model when platform has hard allocation constraint. The only partial results are that, the statement is true for the auction setting, if there is only $1$ bidder or only $1$ item.

## F.2 Discussions on Mechanism Properties

We discuss the three properties we emphasized in Section 2.

- **Potentially exact truthfulness**. The approach should return a mechanism that meets *potentially exact truthfulness*. Since it's hard to verify whether a mechanism is truthful, we often require that any mechanism that can be represented within the parameterized mechanism class should be potentially exact truthful. The *potentially* exact truthfulness means that there is no endogenous factor that makes the mechanism untruthful, *e.g.*, the forced and unreasonable allocation and payment rule. Exogenous factors are acceptable. For example, exogenous factors consist of: floating-point error in computation; irrational behaviors of players when maximizing their utilities; the non-existence of optimal choice when utilities without an upper bound (it can potentially appear in a poor-initialized mechanism). In a nutshell, potentially exact truthfulness requires that players have no reason to complaint about the untruthfulness of the mechanism structure. The regret-minimization-based models (*e.g.*, RegretNet) do not satisfy this property, as the regret can not minimized to be zero. (Dütting et al., 2019; Duan et al., 2022)

- **Full expressive power**. The optimal mechanism can be expressed or approached with arbitrarily small error within the parameterized mechanism class. Since the agnosticism of the optimal mechanism, we often require that any truthful mechanism can be approached with arbitrarily small error within the parameterized mechanism class. The AMA-based model does not satisfy this property, for bounded representative power of AMA model. (Curry et al., 2023; Duan et al., 2024a;b)

- **Efficiency in moderate-size problem**. As the problem size (the number of players and items) increases, there is only polynomial time scale-up for achieving a good-enough mechanism in practice. The menu-based approaches (Wang et al., 2024b; Curry et al., 2023; Shen et al., 2018; Duan et al., 2024a) or programming-based approaches (Wang et al., 2024b; Lavi & Swamy, 2011; Guo et al., 2017) need a discretization on allocation space or type space, indicating an exponentially sample complexity in the worst case.

## G  MORE EXPERIMENTAL DETAILS

### G.1  MORE DETAILS ON BASELINES

#### G.1.1  UM-GEMNET

The input of GemNet (Wang et al., 2024b) with $n$ bidders and $m$ items is $v_{-i} \in \mathbb{R}^{(n-1) \times m}$. We generalize this network's structure to $n = 1$ by the two following approaches:

- Removing the term that penalizes item over-allocation in the original loss function.
- Using $n = 2$ in actual training and divide the testing result by 2. With the previously mentioned change of loss function form, the allocations of the two bidders in the case are independent, thus the two **symmetric** bidders make decisions separately. Hence we obtain a valid result without making large changes to the original GemNet structure.

We use a menu size $K = 300$ if the number of items is less than 5, and $K = 1000$ if otherwise. The network has two hidden linear layers, and the activation function is Leaky-ReLU. The optimizer is Adam, with learning rate $3 \times 10^{-4}$. The softmax temperature when choosing among menu allocations is initialized as 128 and doubles every 500 epochs until the maximum value of 2560. With a minibatch of size $2^{15}$, the training time is 63.55s per 1000 iterations in the 5-item setting and 355.69s per 1000 iterations, respectively (this increase of time is largely due to enlarging the menu from 300 to 1000).

Every 100 epochs, we evaluate the network with $10^5$ samples to check convergence. The converged network is tested on a set of $16 \times 16384 \approx 2.6 \times 10^5$ samples. We observe that the network converges after $\approx 3000$ epochs, and the test performance of $\gg 5000$ (for example , 20000) epochs are not significantly different from that of less than 5000, in a few cases lower.

#### G.1.2  LOTTERY-AMA

We implement the "additive valuation" and "lottery allocations" auction setting of the original method in (Curry et al., 2023). The lottery auctions candidates are generated via item-wise sigmoid instead of item-wise softmax. We still use the two-player training setting in UM-GemNet G.1.1. The learning rate of Adam optimizer is 0.01, and the mechanism is evaluated every 100 epochs with the same validation size as UM-GemNet. We choose the best result among five candidates of lottery allocation size $|A|$ ranging from 2048 to 16384. The training time is $\approx 15$s per 1000 iterations with $|A| = 2048, m = 2$. This method fails to outperform simpler baselines such as item-wise Myerson significantly, when the item number is larger than 5, so the results are omitted.

### G.2  IMPLEMENTATION DETAILS

**Network architecture**   Every network in our experiments is designed as a fully connected network. In the selling experiment, the hidden dimension of neurons $d = 256 \cdot m$ when the network is 1 hidden layer and $d = 32 \cdot m$ when the network is no less than 2 hidden layers. Networks in MoA as well as LSE are always chosen to be with depth 3. PICNN and GroupMax are implemented with depth 1 and 3. When we move to the social planner experiment, the hidden dimension of neurons is decreased to $d = 128 \cdot m$ and the network is fixed to be 1-hidden layer. The positive parameters in neural networks are hardcoded by a softplus function element-wisely: $softplus(x) = \log(1 + \exp(x))$, which maps real numbers onto positive numbers. The convex activation functions are chosen as leaky relu with negative slope 0.01, while other activation functions are chosen as GeLU.

**Training procedures**   We use Adam optimizer and initial learning rate $3 \cdot 10^{-4}$ to optimize all the network parameters and fixed learning rate $3 \cdot 10^{-2}$ to optimize all the non-network parameters. The learning rate of networks is decayed to $\approx 3 \cdot 10^{-6}$ in the training procedure, with each time divided by 2. $\beta$ are chosen $(0.9, 0.9)$ for non-network parameters and $(0.9, 0.999)$ for network parameters.

The penalty weight $\lambda$ in the platform's objective is set with initial value 5, gradually increasing to the maximum value 32. In the first 50 epochs, $\lambda$ increases $\Delta\lambda = 0.02$ in each epoch, and $\Delta\lambda$ increases to 0.03 in later epochs.

When we begin with the training procedure, we first sample $K = 65536$ i.i.d. data, the full procedure only acts on these data. We conduct $T_1 = 300$ epochs for cold start, since both the allocation and the network are initialized and far from optimal. In each epoch in cold start, we train $\theta$ 16 times, train $\boldsymbol{x}$ 8 times and train $\boldsymbol{x}^*$ 32 times. The penalty weight does not change in cold start. After cold start, we continue training $T_2 = 1000$ epochs which we call "hot start". In each epoch in hot start, we train $\theta$ only 4 times, train $\boldsymbol{x}$ 8 times and train $\boldsymbol{x}^*$ only 32 times. When we train $\theta$, we use a random batch with size 2048.

We also validate the models during training. In hot training periods, we save a model per 10 epochs in the first 200 epochs and per 20 epochs in the remaining epochs. We use 65536 sample to validate the model and choose the model with largest platform expected utility on those samples.

**Inference**  In the inference period, we use Adam optimizer with learning rate 0.3 and $\beta = (0.9, 0.9)$. Since the objective function for players is concave, finding the optimal point $\boldsymbol{x}^*$ is a computationally tractable problem. We optimize $\boldsymbol{x}$ with the target of objective function with 500 iterations to simulate the optimal strategies of players. Although it may cause some errors, these errors are at the magnitude $\approx 10^{-10}$, which is sufficiently small that they have negligible errors to the estimation of platform's expected utility.

When testing a model, we use $2^{18} = 262144$ samples to achieve an estimation of expected utility of platform, the standard deviation of estimation error is $\approx 10^{-3}$ times the estimation. Thus it's unlikely that our method outperforms baselines due to random errors.

**Hard-code of** $f_i(\boldsymbol{x}_i; \boldsymbol{t}_{-i}) \le 0$  In MoA model, the *no-buy-no-pay* constraint can be hard-coded as follows,

$$f_i(\boldsymbol{0}; \boldsymbol{t}_{-i}; \theta) = \max_{j \in [K]} b_j(\boldsymbol{t}_{-i}; \theta) \le 0$$
$$b_j(\boldsymbol{t}_{-i}; \theta) \le 0, \forall j.$$

To hardcode $b_j(\boldsymbol{t}_{-i}; \theta) \le 0$, We apply a softplus function to the network output: $\mathrm{softplus}(x) = \log(1 + \exp(x))$, then take the negation.

**Complexity Analysis of these networks**  In this section, we analyze the computational complexity of our approach. Let $n$ and $m$ represent the number of players and items, respectively. The number of training epochs ($300 + 1000 = 1300$), iterations per epoch (approximated as $4 + 8 + 32 = 44$), and the sampling data size (65536) are fixed across different experimental settings. The number of neurons in each network layer is set proportional to $m$, with constant scaling factors (32 for more than 2-layer network and 256 for 1-layer network). We also need to compute the pricing rule for all $n$ players. Consequently, the total computational cost within our framework scales as $nm^2$. Given that the problem description size is $O(nm)$, the training cost scales quadratically with respect to the problem size, which demonstrates the potential to solve large-size problems. The constant coefficient in $O(nm^2)$ approximates as $65536 \times 1300 \times 44 \times 256 \approx 1 \times 10^{12}$.

### G.3  MORE ANALYSIS ABOUT THE RESULTS

**Zero utility of VCG in Table 2**  We derive that when using VCG mechanism, the expected utility of platform might be 0 in some case, which corroborate with the results in Table 2 that VCG utility is 0.

Recall that VCG mechanism maximizes the social welfare: $\sum_{i \in [n]} v_i(\boldsymbol{x}_i; \boldsymbol{t}_i) = \sum_{i \in [n]} \langle \boldsymbol{x}_i, \boldsymbol{t}_i \rangle - \frac{1}{2} \|\boldsymbol{x}_i\|^2$. Let us take the allocation constraints behind for a short time, then the optimal $\boldsymbol{x}_i$ should be chosen at $\boldsymbol{x}_i = \boldsymbol{t}_i$. Therefore, VCG mechanism will result at $\boldsymbol{x}_i = \boldsymbol{t}_i$, then, the platform utility will

become

$$u_0(\boldsymbol{x}; \boldsymbol{t}; \boldsymbol{p}) = \sum_{i \in [n]} v_i(\boldsymbol{x}_i; \boldsymbol{t}_i) - \frac{1}{2} \sum_{j \in [m]} \big( \sum_{i \in [n]} x_{ij} \big)^2$$

$$= \sum_{i \in [n]} \langle \boldsymbol{t}_i, \boldsymbol{t}_i \rangle - \frac{1}{2} \|\boldsymbol{t}_i\|^2 - \frac{1}{2} \left( \sum_{j \in [m]} \sum_{i \in [n]} t_{ij}^2 + \sum_{i_1 \neq i_2} t_{i_1,j} t_{i_2,j} \right)$$

$$= -\frac{1}{2} \sum_{j \in [m]} \sum_{i_1 \neq i_2} t_{i_1,j} t_{i_2,j}$$

By i.i.d. property of $t_{ij}$ and $\mathbb{E}[t_{ij}] = 0$ in each setting, we immediately derive that

$$\mathbb{E}_{\boldsymbol{t}}[u_0(\boldsymbol{x}^{VCG}; \boldsymbol{t}; \boldsymbol{p}^{VCG})] = 0$$

which demonstrates that if allocation constraint always does not bind, then the platform utility of VCG mechanism should be 0. In the case of $t \sim U[-1,1]$ distribution, allocation constraint does not bind indeed.

**The optimal value of some results**   Since the utility function does not depend on $\boldsymbol{p}$, from the above part we know that, if we derive the optimal allocation and allocation constraints do not bind, and additionally the allocation rule $\boldsymbol{x}_i(\boldsymbol{t})$ is implementable (*i.e.*, there is a pricing rule $\boldsymbol{p}_i(\boldsymbol{t})$ that make the mechanism truthful), then, $\boldsymbol{x}_i(\boldsymbol{t})$ must be the optimal allocation that maximizes the platform utility.

Assume the allocation constraints do not bind, by first order condition, we get that,

$$\frac{\partial u_0}{\partial x_i} = 0, \forall i \in [n]$$

It means that

$$\boldsymbol{t}_i - \boldsymbol{x}_i - \sum_{i \in [n]} \boldsymbol{x}_i = 0 \tag{14}$$

Add Equation (14) for all $i$, we know that

$$(n+1) \sum_{i \in [n]} \boldsymbol{x}_i = \sum_{i \in [n]} \boldsymbol{t}_i$$

Taking into Equation (14), we know that the optimal allocation should satisfy:

$$\boldsymbol{x}_i = \frac{n}{n+1} \boldsymbol{t}_i - \frac{1}{n+1} \sum_{j \neq i} \boldsymbol{t}_j \tag{15}$$

In uniform distribution $\boldsymbol{t}_i \in [-1,1]^m$. As long as $n = 2$, we also have that $\boldsymbol{x}_i \in [-1,1]^m$, then allocation constraints do not bind and Equation (15) forms the optimal solution. As long as $n \geq 3$, allocation constraints might bind in some case, and the solution become intriguing.

In the $n = 1$ case, the optimal solution can also be found for all distribution, only by doing a projection on $\boldsymbol{x}_i$ into $[-1,1]^m$. We present a numerical solution of the optimal value in the Gaussian distribution case.

We also note that above-defined $\boldsymbol{x}_i$s are increasing in $\boldsymbol{t}_i$, which make the allocation rule implementable.

**A demonstration of pricing rule**   In this part, we present some pricing rules in different settings. The model we choose in this part is the best model among validation.

PRICING RULE FOR SELLING GOODS TO ONE BUYER    Figure 2 represent the pricing rule learned by PFM-Net with 1-layer GroupMax and 3-layer GroupMax architecture, in the setting of selling $m = 3$ items to one buyer. The $x$-axis represents the allocation on the first item, *i.e.*, $x_1 = x$, while the $y$-axis represents the allocation on the second and third item, *i.e.*, $x_2 = x_3 = y$. The pricing rule is almost piece-wise linear, thus we can approximately take the pricing rule as a bundle mechanism that sells all items at a price $\approx 1.2$, sells one items at a price $\approx 0.8$, and sells two items at the price $\approx 1.4$. Notice that if a player want to buy two items then she must want to buy all items more. Therefore, the mechanism actually do not sell two items, only bundle all items together or sell single item independently. Buyer will get a cheaper average price if she choose to buy the full bundle. This result coincides with the existing finding that optimal mechanism may sometimes bundle all items with a lower price sometimes. The pricing rule of 3-layer GroupMax has a similar regularity. The only difference is that, the price of selling two items is very high in 3-layer GroupMax.

We need to point out that in the characterization of optimal mechanism in selling 3 items (Giannakopoulos & Koutsoupias, 2014), there has small probability that the platform sells exactly 2 items to the player. Our experiments show that if we give up selling exactly 2 items, we will not lose too much.

Figure 3 represent the pricing rule learned by PFM-Net with 1-layer PICNN, 1-layer GroupMax and 3-layer GroupMax architecture, in the setting of selling $m = 20$ items to one buyer. The $x$-axis represents the allocation on the first 10 items, *i.e.*, $x_i = x$ for $1 \leq i \leq 10$, while the $y$-axis represents the allocation on the last 10 items, *i.e.*, $x_i = y$ for $11 \leq i \leq 20$. The pricing rule is almost piecewise linear again. The mode in 1-layer PICNN and 1-layer GroupMax is similar: selling both full bundle and separate bundle, but selling full bundle at a cheaper average price. 3-layer GroupMax again refuses to sell the bundle that consists of exactly 10 items.

PRICING RULE FOR SOCIAL PLANNER OF A MARKET    Figure 4 represent the pricing rule learned by PFM-Net with 1-layer GroupMax architecture, in the setting of social planner with $n = 2$ players and $m = 5$ items. The $x$-axis represents the allocation on the first 2 items, *i.e.*, $x_i = x$ for $1 \leq i \leq 2$, while the $y$-axis represents the allocation on the last 3 items, *i.e.*, $x_i = y$ for $3 \leq i \leq 5$. The pricing rule is non-concave and non-convex, since the pricing rule consists of a convex part $f(\boldsymbol{x}_i; \theta, \boldsymbol{t}_{-i})$ and a regularization part $c_i(\boldsymbol{x}_i)$, which is concave in this setting.

We randomly sample 3 type profiles, and generate the pricing rule given the player 2'th type. We find that the pricing rule for some player highly depends on the other players' types. Consider the case when player 2's type is high on some good, meaning that player 2 is willing the buy the good, it will encourage player 1 to sell the good. Therefore, the social planner want to subsidize player 1 if she really sell the good. The opposite direction is vice versa.

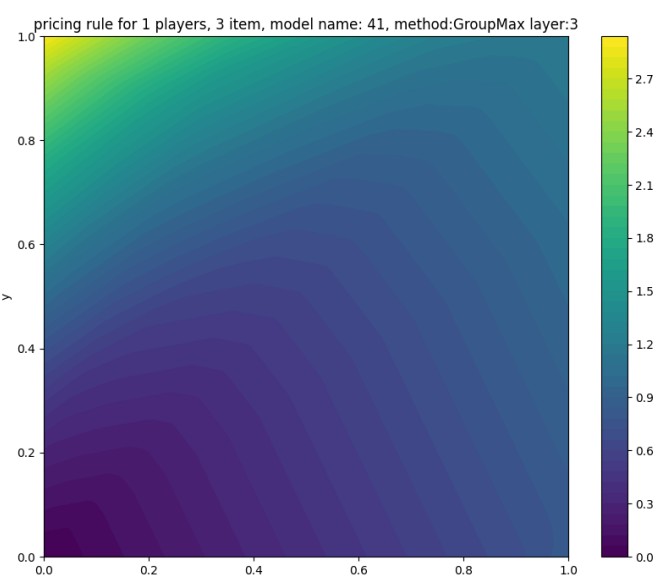

(a) The demonstration of pricing rule learned by 1-layer GroupMax, in the setting of selling $m = 3$ items to one buyer.

(b) The demonstration of pricing rule learned by 3-layer GroupMax, in the setting of selling $m = 3$ items to one buyer.

Figure 2: The demonstration of pricing rule learned by 1-layer GroupMax and 3-layer GroupMax, in the setting of selling $m = 3$ items to one buyer.

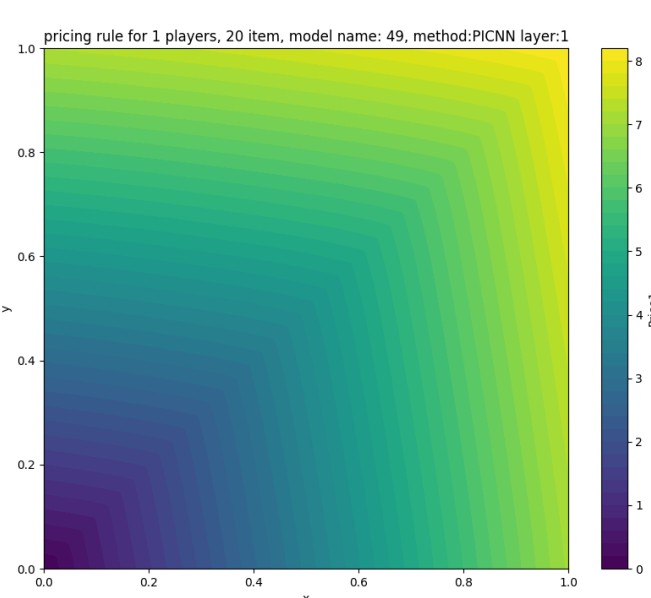

(a) The demonstration of pricing rule learned by 1-layer PICNN, in the setting of selling $m = 20$ items to one buyer.

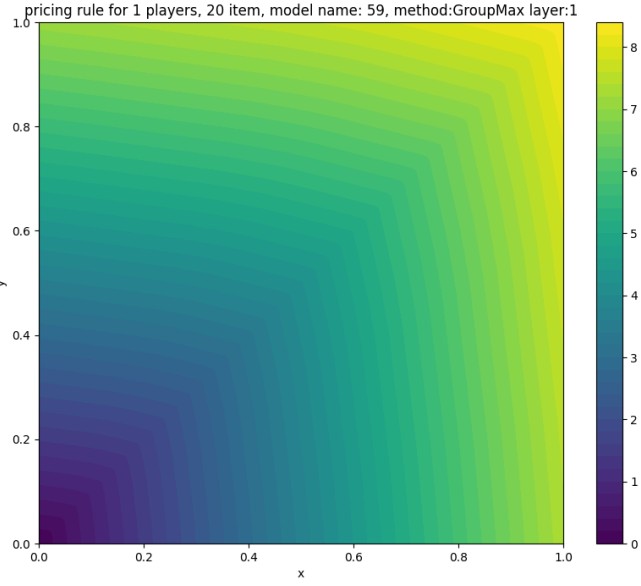

(b) The demonstration of pricing rule learned by 1-layer GroupMax, in the setting of selling $m = 20$ items to one buyer.

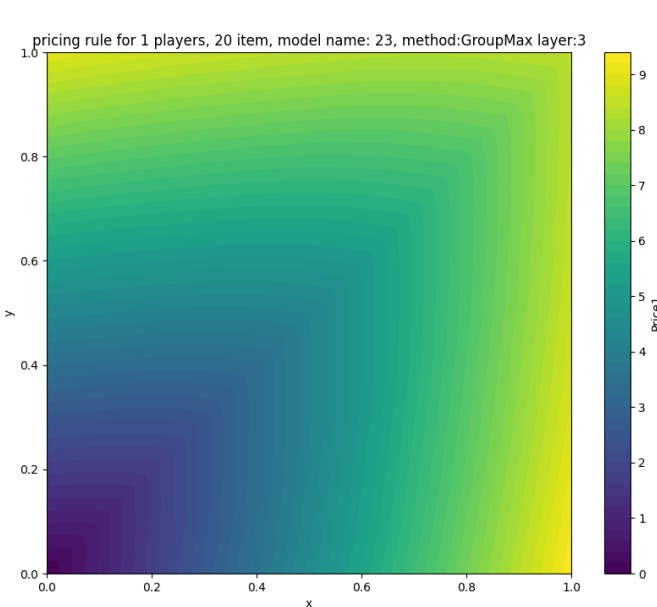

(c) The demonstration of pricing rule learned by 3-layer GroupMax, in the setting of selling $m = 20$ items to one buyer.

Figure 3: The demonstration of pricing rule learned by 1-layer PICNN, 1-layer GroupMax and 3-layer GroupMax, in the setting of selling $m = 20$ items to one buyer.

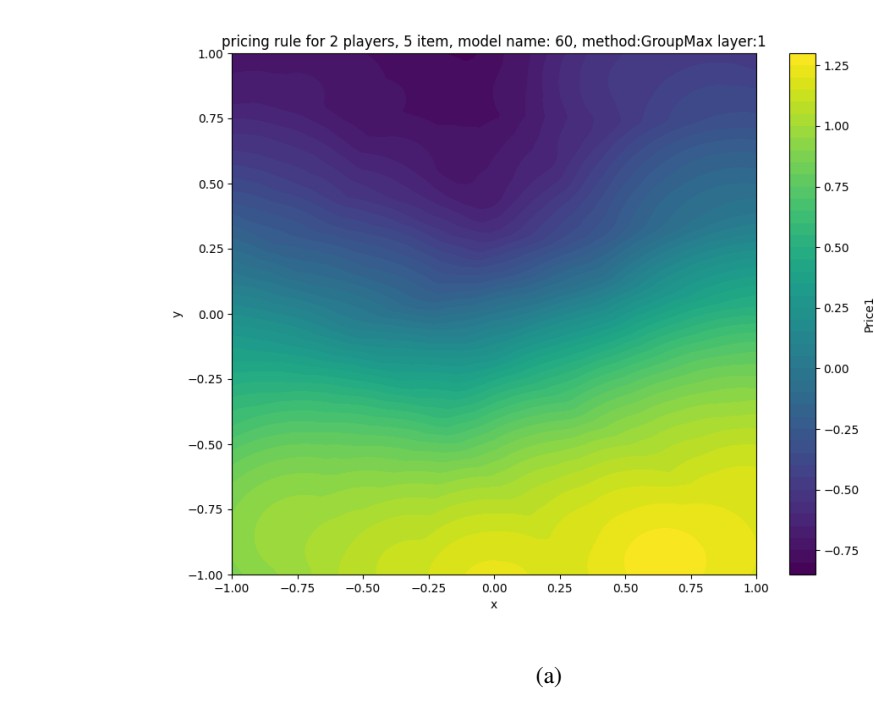

(a)

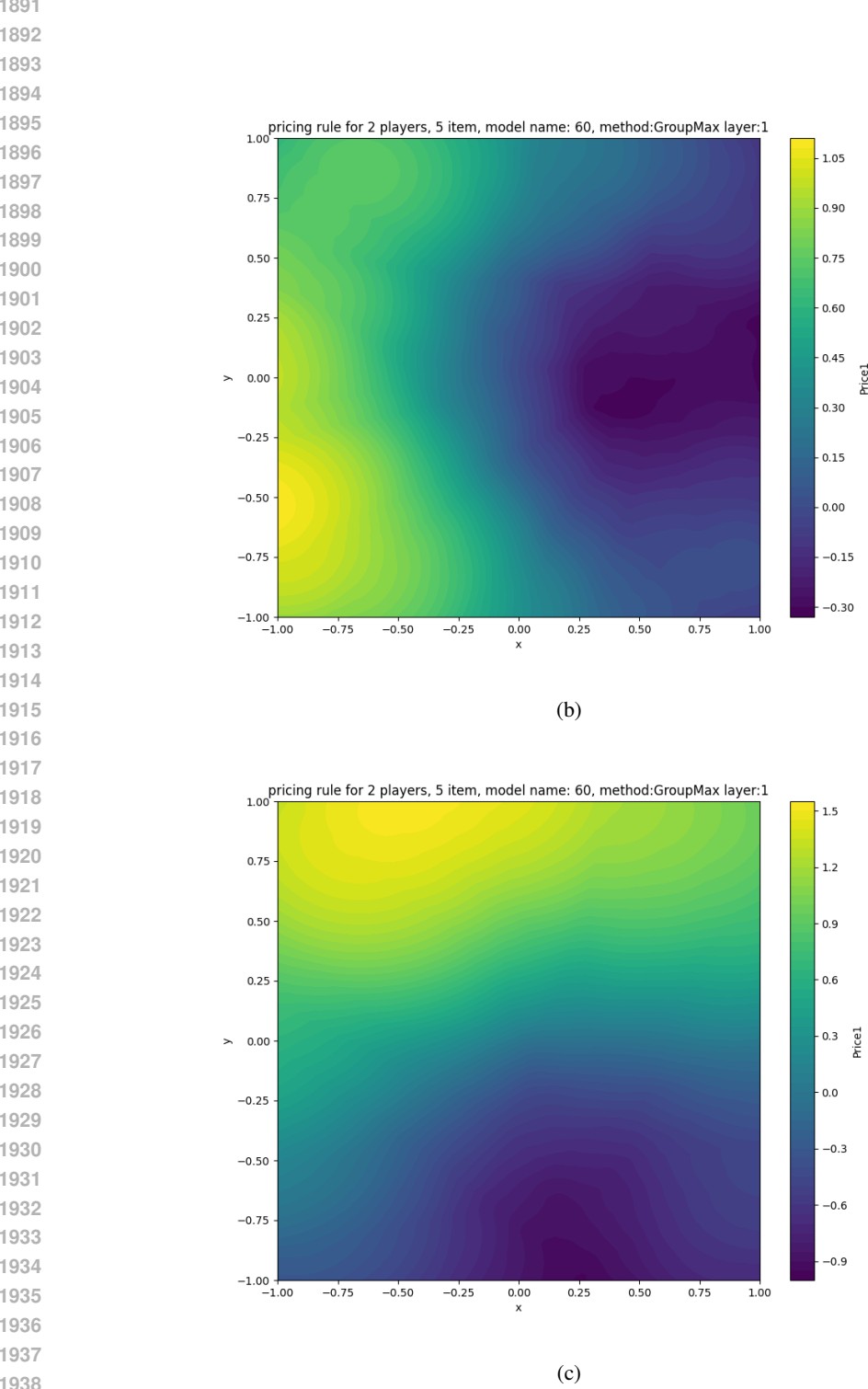

(b)

(c)

Figure 4: The demonstration of pricing rule learned by 1-layer GroupMax, in the setting of social planner with 2 players and 5 items.

