# OpenReview forum: "Learning-based Mechanism Design: Scalable, Truthful, and Continuum Approaches for Utility Maximization"
_ICLR.cc/2025/Conference — Submitted to ICLR 2025_

### Official Review · Reviewer_fyRP · 2024-11-02

**Soundness:** 2
**Presentation:** 1
**Contribution:** 2
**Rating:** 3
**Confidence:** 4

**Summary:**

There has been much recent progress in using neural networks for mechanism design. Current successful approaches are either not fully expressive, are restricted to a single agent, can't guarantee strategyproofness, or require costly postprocessing. The authors of this paper present a new method that is fully expressive and works in general settings, and is strategyproof, yet is claimed not to require costly postprocessing. The key idea is rather than searching for allocation/payment rules as functions of types, they work in the dual space and compute a pricing function over each possible allocation of the mechanism. They show that truthfulness is equivalent to a convexity property of this pricing function. There are many good methods for enforcing convexity of neural networks, so the authors use these to train neural networks representing the pricing function on a couple mechanism design problems, and achieve good performance.

**Strengths:**

The paper tackles a key problem in mechanism design and tries to push things forward in a very creative way. The core idea is clever and original. The authors do have some successful experiments and successfully prove many important mechanism design properties for their method.

**Weaknesses:**

# Presentation

The biggest weakness of this paper is in its presentation.

There are frequent grammar and usage errors, and it might be good to fix them, but these errors don’t harm comprehension, so this is not so important.

However, the organization and structure of the paper is extremely difficult to follow, far beyond the usual problems of space-limited conference papers. The definitions are unusual and non-standard, many points jump around frequently, the proofs are not well-organized, and I find myself trying to guess at what the authors are doing based on my knowledge of mechanism design, rather than learning it from the paper itself.

Overall the presentation is confusing enough that I have a hard time following the paper, even though I completely understand all background work. There are many good aspects to this work but the poor presentation makes it hard to really tell what’s going on.

# Experiments

One of the main exciting things about automated mechanism design is its use in solving the wide-open problem of revenue-maximizing DSIC auction design. There are many auction design benchmark problems in the papers the authors cite (GemNet, RegretNet, AMA) but the authors choose to compare to none of these benchmarks, instead picking only two problems, one of which is less interesting (single buyer) and one of which is not standard.

If the authors could run their method on some of the same benchmarks as in the RegretNet/GemNet papers, and hopefully produce similar plots visualizing the learned mechanisms, it would significantly increase confidence that their method works well. I think any claimed competitor to GemNet must tackle some of those problems shown in the GemNet paper (e.g. recover the Yao auction, or 2x2 uniform additive buyers, at the very least).

# Full Expressiveness and Supply Constraints

The authors claim their method is fully expressive, which seems to be true as far as I can tell. It is also true of GemNet (their main point of comparison).

The main weakness they point out with GemNet is that it requires a costly post-processing step on a discrete grid. The purpose of this post-processing step is to achieve “menu compatibility” — during learning GemNet may choose menus such that when all bidders choose their favorite menu item, some items are oversold, and the post-processing step adjusts prices to prevent this.

So the key point is the post-processing step is only required for problems where menu compatibility is an issue. But in both of the problems studied in the experiments in this paper, menu compatibility would not be an issue — GemNet also is fully expressive and requires no postprocessing!

Although they don’t deal with it in experiments, in principle their method can deal with supply constraints of the sort that show up in auctions. This is discussed quite briefly in section F.1 (I think it belongs in the main body or at least should be mentioned more prominently as it is very important). However there are unresolved issues to make this work in practice. Perhaps these issues can be overcome more efficiently than the GemNet postprocessing, but the paper gives no evidence one way or the other. (Also, side note — equation 13 seems not to “type check” — a vector proj(x, X) is subtracted from a scalar)

Overall as it stands, the authors seem to have pursued a clever idea with some partial success, but this is not a good paper as a paper. I think the authors should significantly rewrite it, as well as pushing experiments much further, and deal more straightforwardly with the issue of handling supply constraints.

**Questions:**

Although this review is somewhat harsh, I do want to give the authors encouragement for pursuing a clever and original idea, and I think in the future this could become a great paper.

The authors are welcome to respond to any points in my review they want to, and to correct any errors or misunderstandings of mine. I would be happy to engage in discussion.

---

> ### Author Response · Authors · 2024-12-02
> **Response to Reviewer fyRP**
>
> We appreciate the reviewer for recognizing our original approach, many useful suggestions as well as encouragement for future study.
> We will address some misunderstandings below.
>
>
> > The definitions are unusual and non-standard
>
> To realize our approach, we indeed need to introduce unusual definitions. But since these definitions are helpful for the understanding of our approach, we believe that they are not a weakness of this work.
>
> > One of the main exciting things about automated mechanism design is its use in solving the wide-open problem of revenue-maximizing DSIC auction design. There are many auction design benchmark problems in the papers the authors cite (GemNet, RegretNet, AMA) but the authors choose to compare to none of these benchmarks.
>
> In this work, we do not focus on auction design (Please refer to our general responses). Therefore, we do not compare multi-player auctions benchmarks.
>
> > I think any claimed competitor to GemNet must tackle some of those problems shown in the GemNet paper
>
> We did not mean to claim a competitor to GemNet, though the menu mechanism approach is inspired by GemNet. Instead, we only focus on the mechanism design problems that do not face the menu compatibility issue, which excludes auction problem. We compare our results with baselines those originate from auction design, mainly because these approaches can be transplanted into our setting with simple modifications.
>
> > The main weakness they point out with GemNet is that it requires a costly post-processing step on a discrete grid.
>
> Actually, we think that (though not pointing out directly) the main weakness of GemNet is that its menu representation needs a discretization on allocation space, which is costly and will result in a suboptimal performance when allocation space becomes high-dimensional. This issue is called "curse-of-dimensionality" in machine learning literature (Please refer to our general responses). Our both experiments also demonstrate this phenomenon.
>
> > Although they don’t deal with it in experiments, in principle their method can deal with supply constraints of the sort that show up in auctions. This is discussed quite briefly in section F.1 (I think it belongs in the main body or at least should be mentioned more prominently as it is very important).
>
> We put this discussion in appendix because our primary focus is not auction setting. We believe that auction is an important problem so we make some attempts to include auctions within our model and try to answer the following question: Can this model express auction problems in an indirect way? Above all, we believe that this discussion is a by-product of this work.
>
> > Also, side note — equation 13 seems not to “type check” — a vector proj(x, X) is subtracted from a scalar
>
> Thanks for pointing out this typo. It should be $ v_0(proj(x, \mathcal{X});t) - M \cdot dist(x, \mathcal{X})$. We will fix this typo in the next version.

---

> > ### Comment · Reviewer_fyRP · 2024-12-02
> > **Thanks for response**
> >
> > Thank you for the considered response. I know that receiving negative reviews can be quite unpleasant. As you continue to work on this paper, I think there is an important point I want to reemphasize.
> >
> > In the non-auction settings with no supply constraints, GemNet does not require any discretization. It will just work without that step. The discretization is only needed to deal with supply constraints. Your approach is different from theirs, but if you are not focusing on auction settings, you cannot claim "no discretization" or avoiding the curse of dimensionality as a particular advantage of PFM-Net compared to existing methods.
> >
> > I would therefore politely encourage you to just lean into the challenge of auctions, expanding on the incomplete ideas in Appendix F.1. Revenue-maximizing auctions are the most immediately exciting application of automated mechanism design anyway.

---

> ### Author Response · Authors · 2024-12-02
> **More Clarifications**
>
> Thank you for your enthusiastic discussion. We will clarify some issues in this response.
>
> * The "curse of dimensionality" of GemNet
>
> We politely mention that the reviewer might have some misunderstanding about what we meant "discretization on allocation space". Though GemNet does not directly discretize the allocation space with menus, the insight shares the commonality with MenuNet, which discretize the allocation space by finite menus for the only one player. We're confident that our understandings about GemNet (its discretization nature on menus) is correct.
>
> To clarify this point, we will take our first experiment as an example ($n=1$ buyer and $m$ items).
> In this case, the network will not receive any input, and GemNet is equal to MenuNet. Therefore, the output of GemNet's network can be seen as separated variables.
> In GemNet's network representation, the output of network is represented as $K$ menus on the $m$-dimensional allocation space. (In their original paper $K$ is chosen as 1000 when $m>5$ and 300 when $m\le 5$)
> We think that such representation intrinsically discretizes the allocation space with $K$ menus.
> However, there have $2^m$ deterministic menus, which becomes exponentially large when $m$ increases. Besides, each deterministic menu among $2^m$ deterministic menus is indispensable in the SJA mechanism (if we translate SJA mechanism into menu mechanism), which is conjectured to be optimal until $m=10$ (please refer to Reviewer sVor's review).
> Therefore, at least in the one player, $U[0,1]^m$ setting, GemNet faces the problem of "curse of dimensionality".
> It is also natural to imagine that the "curse of dimensionality" issue will not disappear in multi-players setting.
>
> * The position of this work
>
> For this work, we did not focus on performing better performance on auctions problem. Our original goal lies in the right structures of automation approaches of mechanism design, which is more theoretical than empirical. We think we are one step closer to the answer, through delving into the functional perspectives rather than point-wise perspectives of menu mechanisms. It's acceptable that we add some assumptions even these assumptions exclude auction settings.
>
> * The future of revenue-maximizing auction
>
> We believe that revenue-maximizing auction is an interesting mechanism design problem, with unique structure (menu compatibility) compared with many other mechanism design settings. Though a post-processing mixed-integer program can resolve menu compatibility issue, we still believe that menu mechanism is not the right approach for efficient auction design.
> Yet it's still hopeful that some approach could perform better on larger settings compared with many current state-of-the-arts.
> Indeed we have some ideas about designing revenue-maximizing auctions under menu compatibility constraints that does not require discretization on allocation space or mixed-integer programming and thus has potential to perform well on large instance. We believe these works will be independent of this submission.

---

> > ### Comment · Reviewer_fyRP · 2024-12-02
> > **Response**
> >
> > Thanks for your clarifying response -- I see now that from your perspective, there are two forms of discretization in GemNet, and you do manage to get rid of one of those.
> >
> > It's not obvious to me that, for example, if you wanted to encode the SJA as the conjugate function rather than the exponentially-large menu, this would actually work much better. However, I suppose there is the possibility that it could, or at least could be easier for purposes of learning. So, good luck!

---

### Official Review · Reviewer_YvjT · 2024-11-03

**Soundness:** 1
**Presentation:** 2
**Contribution:** 2
**Rating:** 3
**Confidence:** 4

**Summary:**

This paper explores deep learning methods applied to mechanism design, focusing on multi-buyer, multi-item scenarios where $ n $ represents the number of buyers and $ m $ the number of items. The primary focus is on menu mechanisms. For example, in a single-buyer context, a menu mechanism is defined by a set $ X \subseteq [0,1]^m $, where each allocation vector $ \vec{x} \in X $ indicates the probability that the buyer will receive each item. This mechanism is coupled with a pricing function $ p(\vec{x}) $, and the buyer selects the allocation vector that maximizes their utility.

The authors attempt to establish that the class of truthful mechanisms is equivalent to the class of menu mechanisms with convex pricing functions (I found this proof somewhat difficult to follow, as I elaborate in the Weaknesses section.)  Building on this theoretical foundation, the authors design a neural architecture. While the main text does not provide details on the architecture or algorithm, the core idea is to use a convex function to represent the payment function, which is optimized during training.

**Strengths:**

The paper's experiments suggest that the authors may be onto something promising with their architectural design. The revenue results in Table 1 indicate strong performance relative to baselines like UM-GemNet, showcasing the potential effectiveness of their approach.

**Weaknesses:**

- I found the proof of the main theoretical result, Theorem 3.4, challenging to follow. This theorem claims that the class of truthful mechanisms is equivalent to the class of menu mechanisms with convex pricing functions, but several parts of the proof were confusing:
  - On line 1018, it’s unclear what is meant by treating $ x_i^d $ and $ p_i^d $ as free variables $ x_i $ and $ p_i $. First, since these are functions, it’s confusing to call them variables, and second, because they are defined by the input mechanism $ M^d $, it’s even more confusing to refer to them as “free” variables.
  - On line 1022, I’m unsure what is meant by saying $ \tilde{u}_i(t) $ is constant with respect to $ x_i $ and $ p_i $. By definition, $ \tilde{u}_i(t) $ depends on $ x_i^d $ and $ p_i^d $, so it doesn’t appear to be constant with respect to these terms.
  - In Equation (5), the supremum is taken over $ t_i $, but the line before mentions $ p_i $ should be minimized. The connection between this minimization and the supremum in Equation (5) is unclear.
  - The paragraph titled “Prove the first statement” on line 1049 is also difficult to interpret. Since $ \tilde{u}_i(t) $ is a function of $ x_i^d $ and $ p_i^d $, whether $ \tilde{u}_i(t) $ is convex should depend on the properties of $ x_i^d $ and $ p_i^d $ (e.g., whether or not they themselves convexity).
- Section 4 would benefit from more information on the algorithm.
- In terms of experiments, prior work (e.g., UM-GemNet) evaluates performance on a wider range of distributions beyond $ U([0,1]) $. This paper should expand its set of benchmarks to allow for a more comprehensive comparison.
- There are also numerous grammatical issues throughout the paper. I recommend using a tool like Grammarly to identify and correct these. Lastly, it’s advisable to avoid terms like “ingenious” in the abstract when describing one’s own method.

**Questions:**

Could you please address my confusions regarding the proof of Theorem 3.4?

---

> ### Author Response · Authors · 2024-12-02
> **Response to Reviewer YvjT**
>
> We appreciate the reviewer for suggestions on the organization of Section 4, more experiments as well as grammatical errors. We will address the confusions about the proof below.
>
> > On line 1018, it’s unclear what is meant by treating $x^d_i$ and $p^d_i$ as free variables $x_i$ and $p_i$.
> >
> > On line 1022, I’m unsure what is meant by saying $\tilde{u}_i(t)$ is constant with respect to $x_i$ and $p_i$. By definition, $\tilde{u}_i(t)$ depends on $x^d_i$ and $p^d_i$, so it doesn’t appear to be constant with respect to these terms.
>
> The statements between line 1018 to line 1029 are only the insight why we construct $p^f_i(x_i, t_{i})$ as the form in line 1032. Removing them does not affect the correctness of this proof.
>
> The statements actually assume that we can replace $x^d_i$, $p^d_i$ with $x_i$, $p_i$, respectively.
>
> > In Equation (5), the supreme is taken over $t_i$, but the line before mentions $p_i$ should be minimized.
>
> Since we want the inequality in Equation (4) always hold and take equality sometimes, then, we should choose $p_i$ such that $p_i$ is exactly the form in line 1032. The form of $p_i$ in line 1032 is also the minimum value such that Equation (4) always hold, because if $p_i$ get smaller, then by definition of supreme, there will be some $t_i$ that violates the inequality in Equation (4).
>
> > The paragraph titled “Prove the first statement” on line 1049 is also difficult to interpret. Since $\tilde{u}_i(t)$ is a function of $x^d_i$ and $p^d_i$, whether $\tilde{u}_i(t)$ is convex should depend on the properties of $x^d_i$ and $p^d_i$ (e.g., whether or not they themselves convexity).
>
> We do not need to prove $\tilde{u} _i(t)$ is convex. What we should prove is that $p^f _i(x _i;t _{-i})$ is convex w.r.t. $x_i$. Our original proof says that, since $p^f _i(x _i;t _{-i})$ has the form in Equation (5), which is exactly the Fenchel conjugate form, then it is convex w.r.t. $x_i$ by nature.

---

### Official Review · Reviewer_sVor · 2024-11-04

**Soundness:** 2
**Presentation:** 2
**Contribution:** 2
**Rating:** 3
**Confidence:** 5

**Summary:**

There has been a lot of recent research on designing revenue-optimal, strategy-proof auctions through the use of neural networks and machine learning tools. However, existing approaches often fall short of meeting all desired properties: exact truthfulness, expressiveness, and efficiency. For instance, RegretNet does not ensure exact truthfulness, AMA mechanisms lack expressive power, and MenuNet can be computationally inefficient. This paper presents PFM-Net, a framework designed to address all three objectives through a learning-based approach.

The authors propose a full-menu mechanism that uses neural networks to parameterize pricing functions—determining how much agents are charged for specific bundles based on their valuations. This framework incorporates insights from economic theory, such as agent independence, convexity and monotonicity, to achieve incentive compatibility and a no-buy-no-pay rule to satisfy individual rationality. The optimization is an alternating process: first, allocations are optimized for the players given fixed pricing functions; then, the neural network parameters are adjusted to optimize revenue for the auctioneer.

The framework is initially evaluated in a single-bidder, multiple-goods setting. It is adaptable to other objectives, such as social welfare maximization. To demonstrate this flexibility, the authors include an experiment with a social planner setting involving multiple agents and multiple goods.

**Strengths:**

S1. PFM-Net leverages insights from economic theory, including agent independence, convexity, and monotonicity, to ensure incentive compatibility and no-buy no-pay rule to satisfy individual rationality. These designs seem to be an improvement over architectures presented in RegretNet

S2. Avoidance of Explicit Menu Enumeration
Traditional menu-based mechanisms often require enumerating all possible menu options, which can be computationally prohibitive, especially as the number of items grows. For instance, even a deterministic mechanism for a single buyer with $m$ items would need $2^m$ menu options, creating scalability challenges. PFM-Net, however, avoids this by not requiring explicit enumeration of the menu. For each auction instance, it optimizes the agent's objective directly based on the pricing functions. This means allocations are determined dynamically by maximizing the agent’s utility under the current pricing function, allowing the model to handle large settings without incurring the overhead of menu enumeration.

**Weaknesses:**

W1. Missing Baselines
RochetNet, the current state-of-the-art for single-buyer settings, should be included as a benchmark, as other methods generally reduce to RochetNet in this context, making it a sufficient point of comparison. Additionally, the optimal mechanism for up to six items is given by the SJA mechanism (referenced in the paper as Giannakopoulos and Koutsoupias). Please include this under OPT for $S_5$. This mechanism can be extended for larger m using a recursive formula, with results available up to m = 10 in the RegretNet paper, where it is also conjectured to be optimal. This makes SJA an essential baseline for evaluating the proposed approach.

Moreover, it would be interesting to test the model’s performance in a setting with a single additive bidder and two items, where the bidder's values are independently drawn from a Beta distribution (α=1, β=2). Prior work [2] has shown that the optimal mechanism in this setup involves an infinitely sized menu, providing a valuable test case. Additionally, including settings where randomization is essential would show how well this approach performs for non-deterministic settings.

W2. Lack of Moderate/Large-Scale Experiments
The paper currently lacks experiments involving moderate to large-scale settings. RegretNet already performs well with very low regret for the small scale settings shown in the paper. For smaller settings at least, one could consider Regretnet to be potentially exactly truthful. To fully demonstrate PFM-Net's advantages over regretnet, it would be beneficial to include tests with multiple agents and items (e.g., n,m≥2)

W3. Writing and Clarity
The writing is generally clear and accessible until Section 4. I found myself frequently switching between the appendix and the main paper to fully understand the methodology. Including the learning algorithm or a pseudo-code in the main paper would improve readability. Additionally, clearly noting in the main text when specific technical details, such as the handling of over-allocations, are explained in the appendix would be helpful as well.

---

### References
[1] Yiannis Giannakopoulos and Elias Koutsoupias. Duality and optimality of auctions for uniform
distributions. In Proceedings of the fifteenth ACM conference on Economics and computation,
pp. 259–276, 2014.

[2] Daskalakis, C., Deckelbaum, A., and Tzamos, C. (2017). Strong duality for a multiple-good monopolist. Econometrica, 85:735–767

**Questions:**

This approach shares notable similarities with RegretNet. Rather than having separate networks for allocation and payment, PFM-Net combines these into a single payment network with hardcoded constraints like convexity and monotonicity. The training process is also comparable: the proposed approach alternates between computing the allocation (the analogous step is finding the misreport in regretnet) and optimizing the payment function (updating the weights of RegretNet). Equation 11 is the also the same as RegretNet's objective (with the missing L2 penalty term).

For larger settings, it’s likely that PFM-Net would encounter similar issues to RegretNet with gradient-based allocation computation. Testing the proposed approach's exact violation (i.e., the term involving ReLU in Equation 11) at test time, preferably with multiple initializations of x, would be informative. For smaller settings, RegretNet incurs very low regret (and recovers optimal solutions wherever known). For larger configurations, further evaluation is needed to verify that PFM-Net’s allocation computation can overcome these scaling challenges that RegretNet faces in computing the misreports.

---

> ### Author Response · Authors · 2024-12-02
> **Response to Reviewer sVor**
>
> We appreciate the reviewer for suggestions on the missing baselines and beta distribution settings. We will address some misunderstandings in this response.
>
> > About the problem studied in this paper
>
> The reviewer seems to misunderstand that our paper focus on auction design. Actually, auction is not our focus, and unfortunately our model can not explicitly express auction problems (please refer to our general responses), and that's the reason why we do not conduct multi-bidder auction experiments
>
> > To fully demonstrate PFM-Net's advantages over regretnet, it would be beneficial to include tests with multiple agents and items (e.g., n,m≥2)
>
> Our second experiments consider a multi-bidder, multi-item setting, though this setting is not a standard auction setting.
>
> > The similarity between PFM-Net and RegretNet
>
> The reviewer thinks PFM-Net shares similarity with RegretNet. Actually, PFM-Net is far different from RegretNet due to following reasons.
>
> * RegretNet is intrinsically untruthful (because the training process needs to minimize regret), while PFM-Net is intrinsically truthful (because PFM-Net is menu-based, and menu mechanism is intrinsically truthful). (Please refer to our general responses)
> * RegretNet faces the problem of instability in finding misreport when the problem size becomes large, which we believe the main reason is that finding misreport is a high-dimensional non-convex optimization problem for RegretNet. In PFM-Net, the pricing rule is designated to be convex, so in the "computing the allocation" step, we face a convex optimization problem, and thus can be efficiently computed.
>
> > The truthfulness of PFM-Net
>
> The reviewer seems to misunderstand the *potential exact truthfulness* of PFM-Net, and think PFM-Net might be untruthful in some cases.
>
> The word *potential* arise from the rationality assumption about computing allocation: we assume that the mechanism/player has the ability to choose the allocation that maximizes players' utilities. (We need this assumption because the allocation set is an continuous set.) Under this reasonable assumption, PFM-Net is exact truthful.
>
> In experiments, we find the optimal allocation by convex optimization that maximizes players' utilities. In this sense, we believe that testing the regret of PFM-Net is unnecessary since it is already truthful.

---

> ### Comment · Reviewer_sVor · 2024-12-02
>
> Thank you for the clarification. While your explanation resolves my concerns regarding the similarities and differences between this proposed approach and RegretNet, my other issues remain unaddressed (hence my decision to maintain my scores). To reiterate:
>
> W1: The baselines are still missing in the paper!
>
> W2: While it’s acceptable that your focus is not on multi-buyer auction settings, this needs to be clearly stated. However, note that if your approach only works for a specific mechanism design problem and cannot be extended to auction design, the contributions become quite narrow.
>
> W3: The paper's writing could be significantly improved.
>
> For these reasons, I will maintain my scores.
>
> That said, I think this is a great idea overall. The biggest strength of the proposed approach is that it eliminates the need to enumerate and select from all possible menu options (as required by methods like RochetNet, MenuNet, AMANet, GemNet, etc.) and instead computes the allocation by solving a convex optimization problem. To strengthen this paper, I recommend the authors the following:
>
> 1. Focus on the strengths for the single-item setting. For example, in cases with \(k > 20\) items, enumerating the deterministic menu (\(2^k\)) becomes intractable. Your approach has the potential to excel here.
>
> 2.  Improve clarity and readability to make the paper more accessible to the audience.
>
> 3. Since there isn’t new theory, focus on experiments to demonstrate your approach’s superiority. For instance:
>    - Include experiments showing that your approach can recover optimal solutions for known settings (both deterministic and with lotteries).
>    - Provide experiments where your approach outperforms existing methods.

---

> > ### Author Response · Authors · 2024-12-03
> > **Thank you for Responses**
> >
> > Thank you for your further suggestions and encouragements about improving this work! We understand that there seems to be limitations on the experiments and writings in this version and thus this paper could not worth an acceptance. We will improve these aspects in the next version.
> >
> > Side note, we think the functional perspectives of menu mechanisms and its full-expressiveness characterizations are a new theory in automated mechanism design. This observation allows us to utilize universal approximators of convex functions to parameterize the menu mechanism instead of directly enumerating menus, without worrying that the optimal mechanism would be lost.

---

### Official Review · Reviewer_ZBuV · 2024-11-08

**Soundness:** 2
**Presentation:** 1
**Contribution:** 1
**Rating:** 1
**Confidence:** 5

**Summary:**

This paper uses the MenuNet (Shen et al., 2018) idea to optimize a general quasi-linear objective over multiple players and multiple items. In particular, the feasible allocation set X is the product of the feasible allocation set for each individual player i, which implies that there is no constraint on the total supply of each item (each item could potentially allocate to multiple players).

The results are compared with other mechanisms empirically.

**Strengths:**

Unless I misunderstood the feasible allocation set, I didn’t see much strength.

**Weaknesses:**

The problem being solved in this paper is a trivial extension of the MenuNet (Shen et al., 2018) to multiplayer setting with individual-wise allocation constraints. The key challenge of extending MenuNet to general multi-bidder setting is to handle the feasibility constraint properly (i.e., each item can only be allocated to at most one bidder).

This challenge is referred to as “menu compatibility” and first solved by GemNet (Wang et al, 2024b), who solve the compatibility issue through a combination of a price adjustment and MIP (mixed integer program).

This paper, however, drops the only challenge of generalizing MenuNet to the multi-bidder setting. So I cannot see any real contribution to the literature (unless I misunderstood this part).

**Questions:**

Please properly mention the key result of MenuNet in your second paragraph, and explicitly compare your approach with theirs.

---

> ### Author Response · Authors · 2024-12-02
> **Response to Reviewer ZBuV**
>
> Thanks for the reviewer's comments. We will address some misunderstandings of this paper.
>
> > The problem being solved in this paper is a trivial extension of the MenuNet (Shen et al., 2018) to multiplayer setting with individual-wise allocation constraints.
>
> The extension is not trivial. Even in the 1-buyer case, PFM-Net is far different from and performs better than MenuNet. (Please refer to our general responses.)
>
> > This paper, however, drops the only challenge of generalizing MenuNet to the multi-bidder setting
>
> We indeed do not resolve the challenge of menu compatibility, but menu compatibility is not the focus of this paper.
>
> However, there is one challenge that is missed in all automated mechanism design literature, which is called "curse of dimensionality''(Please refer to our general responses). MenuNet and GemNet suffer from curse of dimensionality inevitably, while our approach can escape from it. Both experiments validate this statement.
>
> > Please properly mention the key result of MenuNet in your second paragraph, and explicitly compare your approach with theirs.
>
> GemNet, as an extension of MenuNet, degenerates to MenuNet when there is only one player. Therefore we believe that comparing with MenuNet separately is unnecessary, as long as we compare with GemNet.

---

### Author Response · Authors · 2024-12-02
**An Overall Clarification to All Reviewers**

Dear all Reviewers,

We have carefully read all the reviews. We thank all the reviewers for their efforts and valuable suggestions.

We apologize that our organizations made reviewers confused, and there seems to be some misunderstandings in the reviews. Please allow us make some clarifications that demonstrates the value of this work. Besides, we also address the concerns of reviewers in separate responses.

* About the problems studied in this paper:

In line 117-118, we wrote, "Denote $\mathcal{X} = \prod_{i \in [n]} \mathcal{X}_i$ as the possible allocation set. Note that this model implicitly means that the constraints are endogenous from players, rather exogenous from the platform."

This paper makes a model assumption, $\mathcal{X} = \prod_{i \in [n]} \mathcal{X}_i$, which means that we do not need to consider menu compatibility issue in our model. Apparently our model can not express auctions by this assumption, and we believe this point is the only limitation of this work.

We will emphasize that the model assumes $\mathcal{X} = \prod_{i \in [n]} \mathcal{X}_i$ and no menu compatibility issue will arise in next version.

* About the Contribution of the Paper:

The main contribution of PFM-Net compared with existing methods is that we **incorporates network-based functional approximation to the pricing rule architecture** (within menu mechanism), which is an original study in the field of automated mechanism design. This improvement helps menu-based mechanism escaping "curse of dimensionality"[1], and shows effective results in experiments (even in the simple 1-player setting).

[1] Richard Bellman. Dynamic programming. science, 153(3731):34–37, 1966.

* What's the advantage of PFM-Net over Regret-based methods and AMA-based methods?

Regret-based methods suffer from intrinsic untruthfulness, and AMA-based methods suffer from limited expressiveness. Even though the metric of untruthfulness (called regret in RegretNet) can be minimized during training, it's impossible that the regret decreases to $0$. On other side, the performance of Regret-based methods often exceeds the theoretical OPT in many experiments, making such results meaningless.

Menu-based methods are intrinsically truthful and hold full expressive power. PFM-Net is a universal approximator of the equivalent class of truthful mechanisms, which demonstrates that PFM-Net is also intrinsically truthful and hold full expressive power.

* Why does not this paper consider menu compatibility as GemNet does?

GemNet (Wang et al. 2024b) utilizes mixed-integer programming to resolve the menu compatibility issue.
We believe that the mixed-integer programming is a general tool to transform any mechanism to be menu-compatible (even without the training step in GemNet). However, we also believe that mixed-integer programming sheds no light on the structure of automated mechanism design.
Our results show that, if we can assume more $\mathcal{X} = \prod \mathcal{X}_i$, then menu mechanisms with parameterized pricing rule seems to be the right perspective to automate truthful mechanism design.

* About the Limitations of PFM-Net:

We believe that the only limitation of this work is that our model can not directly express multi-bidder auctions, and that's why we do not conduct multi-bidder auction experiments. We also discuss some other alternative solution concepts in appendix F.1, which indicates that our model can express multi-bidder auctions in an indirect way.

However, we argue that our model can express many mechanism design problems outside traditional auctions, such as reverse auctions, digital goods auctions and all settings in our experiments, as long as they do not face the menu compatibility issue. These mechanism design problems are also worth studying, thus we believe that this limitation is not fatal.

---

As is pointed out by many reviewers, this paper still face some other issues including grammars, the not easy-to-understand organizations as well as immature experiments. Therefore, we will withdraw this version after the discussion period and improve this work before next submission.

---

### Meta-Review · Area_Chair_DYSL · 2024-12-19

**Metareview:**

This paper studies automated mechanism design and proposes a full-menu mechanism that uses neural networks to parameterize pricing functions. The paper presents some interesting ideas such as avoiding explicit menu enumeration. However, all reviewers recommend rejection, sharing concerns that the current presentation is hard to follow and the experiments are not convincing. I agree with the reviewers and recommend rejection.

**Additional Comments On Reviewer Discussion:**

The reviewers shared concerns about presentation and the experiments. The response did not fully address the questions and the reviewers maintained their scores.

---

### Decision · Program_Chairs · 2025-01-22

Reject